# Prior Specification for Exposure-based Bayesian Matrix Factorization

**Zicong Zhu**  *zczhu@is.s.u-tokyo.ac.jp*
*Department of Computer Science*
*The University of Tokyo*

**Issei Sato**  *sato@g.ecc.u-tokyo.ac.jp*
*Department of Computer Science*
*The University of Tokyo*

**Reviewed on OpenReview:** *https://openreview.net/forum?id=o5R4Hv9XqC*

## Abstract

The rapid development of the Internet has resulted in a surge of information, particularly with the rise of recommender systems (RSs). One of the most significant challenges facing existing RS models is data sparsity. To address problems related to sparse data, Bayesian models have been applied to RS systems because of their effectiveness with small sample sizes. However, the performance of Bayesian models is heavily influenced by the choice of prior distributions and hyperparameters. Recent research has introduced an analytical method for specifying prior distributions in generic Bayesian models. The major concept is a statistical technique called Prior Predictive Matching (PPM), which optimizes hyperparameters by aligning virtual statistics generated by the prior with observed data. This approach aims to reduce the need for repeated and costly posterior inference and enhance overall Bayesian model performance. However, our evaluation of this theoretical method reveals considerable deviations in prior specification estimates as data sparsity increases. In this study, we present an enhanced method for specifying priors in Bayesian matrix factorization models. We improve the estimators by implementing an exposure-based model to better simulate data scarcity. Our method demonstrates significant accuracy improvements in hyperparameter estimation during synthetic experiments. We also explore the feasibility of applying this method to real-world datasets and provide insights into how the model's behavior adapts to varying levels of data sparsity.

## 1 Introduction

Recommender systems are defined as information filtering and retrieval systems (Belkin & Croft, 1992), playing a significant role in our daily lives. The goal of these systems is to make accurate and satisfactory recommendations for users based on a limited number of observations. These observations typically consist of interactions between users and items, such as user ratings and item clicks. Over the past few decades, latent factor models have had a profound impact on the field of recommender systems. These models assume that users' preferences are determined by a set of latent factors (Ricci et al., 2010). The common approach in these models involves representing observed ratings as the inner product of user and item factor matrices. Matrix Factorization (MF) refers to a group of algorithms widely used to learn these latent factor models by decomposing the observed matrix into the product of two matrices with lower dimensions (Koren et al., 2009). Various types of MF algorithms have been developed for different scenarios. For instance, Funk (2006) proposed Funk SVD, which demonstrated outstanding performance in the Netflix Prize contest. Paterek (2007) built upon Simon's work by incorporating user and item biases, achieving higher prediction accuracy in the same contest. Additionally, Non-negative Matrix Factorization (NMF) has been applied to recommender systems when the non-negativity of the resulting matrices is required (Luo et al., 2014).

Due to the increasing demands of handling sparse and imbalanced real-world datasets, Bayesian models have attracted significant attention from researchers. Several novel models have been proposed to address the matrix factorization (MF) problem in a probabilistic manner, including Probabilistic MF (PMF) (Mnih & Salakhutdinov, 2007), Bayesian Probabilistic MF (BPMF) (Salakhutdinov & Mnih, 2008), and Generalized Probabilistic MF (GPMF) (Shan & Banerjee, 2010). Compared to classical deterministic models, Bayesian models naturally integrate prior information with observed data within a rigorous framework, enabling them to effectively handle small sample sizes (Van de Schoot et al., 2014). Although Bayesian models offer greater flexibility in dealing with sparse and imbalanced data, as well as comprehensive expressions of uncertainty and convenient updating rules, proper prior specification remains a critical issue that directly affects their performance (da Silva et al., 2023). Unfortunately, there is no standard statistical method for selecting prior distributions or specifying priors for all unknown hyperparameters. In practical implementations, optimal hyperparameters are typically determined through Bayesian optimization (Snoek et al., 2012) or cross-validation (Vehtari et al., 2017). However, these evaluation methods require repeatedly performing computationally expensive posterior inferences across all possible hyperparameter sets. Prior specification offers advantages, but it also faces challenges such as computational efficiency and the trade-off between model complexity and the interpretability of latent factors.

Recently, da Silva et al. (2023) proposed a state-of-the-art prior specification method for Bayesian MF based on the idea of prior predictive matching. This method allows for the estimation of appropriate hyperparameters, including the number of latent factors, without requiring redundant posterior inference. In their empirical experiments using synthetic datasets, they found that the relative error between estimated hyperparameters and true settings is quite small when the dataset is nearly dense (with sparsity less than 10%). However, this error grows rapidly as sparsity increases. These negative results, heavily affected by data sparsity, have hindered further research and practical applications, as most real-world datasets are more than 99% sparse. We also conducted additional experiments on the real-world datasets (Harper & Konstan, 2015) for the uncontrollable prior seetings and attached the results in Appendix C.

In this study, we outline our contributions as follows:

- We evaluated a state-of-the-art prior specification method for Bayesian Matrix Factorization (MF) under varying levels of data sparsity and analyzed the potential reasons for inaccurate hyperparameter estimation.

- We incorporated the concept of the exposure-based model (Liang et al., 2016) to capture the effects of data sparsity and derive improved hyperparameter estimators.

- We conducted experiments on synthetic datasets, demonstrating that our new estimators outperform existing methods, especially as the dataset becomes sparser.

- We investigated how our estimators respond to changes in data sparsity and analyzed the features of estimators. We discussed the limitations of the conventional Probabilistic Matrix Factorization (PMF) model.

## 2 Preliminaries

### 2.1 Bayesian Matrix Factorization

**Fundamental notations** Let $Y \in \mathbb{R}^{I \times J}$ denote the observed interaction matrix with $I$ users and $J$ items, $P \in \mathbb{R}^{K \times I}$ refer to the latent matrix of the user, and $Q \in \mathbb{R}^{K \times J}$ refer to the latent matrix of the item. Denote by $K$ the number of latent factors, which is a hyperparameter for latent factor models and it holds the condition that $K \ll \min\{I, J\}$. For a particular position with user $i$ and item $j$, $p_i \in \mathbb{R}^{K \times 1}$ denotes the user latent vector and $q_j \in \mathbb{R}^{K \times 1}$ denotes the item latent vector. The corresponding preference score from user $i$ to item $j$ is given by the inner product that $Y_{ij} = p_i^\top q_j$.

**Generic Bayesian MF**  A generalized template model for Bayesian MF can be described in the following form:

$$p_{ik} \sim F(\mu_p, \sigma_p^2), \quad q_{jk} \sim F(\mu_q, \sigma_q^2),$$
$$Y_{ij} \sim F_Y\left(\sum_{k=1}^K p_{ik} q_{jk}, \sigma^2\right) \quad \text{with } \mathbb{E}[Y_{ij}] = \sum_{k=1}^K p_{ik} q_{jk}. \tag{1}$$

Figure 5 shows the model structure of the generic Bayesian MF. The user latent variable of the vector $p_i$ follows priors $F(\mu_p, \sigma_p^2)$, the item latent variable vector $q_j$ follows priors $F(\mu_q, \sigma_q^2)$, and the hyperparameters can be specified separately. $F_Y$ is the observation model. The inner product of the latent vectors of the user and the item is no longer used directly as predictions, but the expectation of the observation model with natural noise $\sigma$. The observation model also implies the capability of the Bayesian model to provide results with uncertainty.

**Particular selections of parametric distribution**  Different selections of the prior distribution and the observation model lead to different Bayesian MF models. We introduce several examples:

- Mnih & Salakhutdinov (2007) proposed the classic Probabilistic MF (PMF). This model selects zero-mean Gaussian priors for latent features and a Gaussian distribution for observations:

$$F(\mu_p, \sigma_p^2) = \mathcal{N}(\mu_p = 0, \sigma_p^2), \quad F(\mu_q, \sigma_q^2) = \mathcal{N}(\mu_q = 0, \sigma_q^2), \quad F_Y = \mathcal{N}. \tag{2}$$

- Both Cemgil (2009) and Gopalan et al. (2014) contributed to Poisson MF. This model selects Gamma priors for latent features and a Poisson distribution for observations:

$$F(\mu_p, \sigma_p^2) = \text{Gamma}(a, b), \quad F(\mu_q, \sigma_q^2) = \text{Gamma}(c, d), \quad F_Y = \text{Poisson}. \tag{3}$$

  $a/c$ are the *shape* parameters and $b/d$ are the *rate* parameters of Gamma distributions. Therefore we have the following equalities:

$$\mu_p = \frac{a}{b}, \quad \sigma_p^2 = \frac{a}{b^2}, \quad \mu_q = \frac{c}{d}, \quad \sigma_q^2 = \frac{c}{d^2}.$$

- Salakhutdinov & Mnih (2008) proposed the Bayesian Probabilistic MF (BPMF). This model selects Gaussian-Wishart priors for the prior distribution $F$. The hyperparameters $\mu_p$, $\sigma_p$, $\mu_q$, $\sigma_q$, are controlled by additional (higher-level) priors. This type of model is known as a hierarchical Bayesian model.

## 2.2  Exposure-based Model

**Missing-Not-At-Random (MNAR)**  In this research, we treat the term *data sparsity problem* as a synonym for the term *missing data problem*. According to Rubin (1976), missing data problems can be classified into three categories: Missing-Completely-At-Random (MCAR), Missing-At-Random (MAR), and Missing-Not-At-Random (MNAR). For example, da Silva et al. (2023) address the data sparsity of a synthetic dataset by using a Bernoulli random variable mask. This typically represents an MNAR case, indicating that the probability of data being missing depends on unknown factors. Additionally, making MNAR assumptions helps us gain insights for modeling data sparsity.

**Exposure MF (ExpoMF)**  The positive-unlabeled problem refers to the trivial consideration of missing implicit feedback as a negative response (Jannach et al., 2018). This treatment can be inappropriate because the items are possibly not exposed instead of disliked by the users. Liang et al. (2016) proposed this probabilistic model with an exposure variable to solve this problem. The exposure variable modernizes the

missing pattern of data in a not-random manner. The ExpoMF is defined as follows:

$$\text{Latent Factors:} \qquad p_i \sim \mathcal{N}(0, \sigma_p^2 \mathbf{I}), \quad q_j \sim \mathcal{N}(0, \sigma_p^2 \mathbf{I}).$$

$$\text{Exposure Mechanism:} \quad O_{ij} \sim \text{Bern}(\theta_{ij}). \tag{4}$$

$$\text{Observed Feedback:} \qquad Y_{ij} \mid O_{ij} = 1 \sim \mathcal{N}(p_i^\top q_j, \sigma^2), \quad Y_{ij} \mid O_{ij} = 0 \sim \delta_0.$$

$Y_{ij} \in \{0, 1\}$, denotes the implicit feedback, and $\delta_0$ denotes a point mass at 0, for example, $Pr(Y_{ij} = 0 | O_{ij} = 0) = 1$. $\theta_{ij}$ is the prior probability of exposure controlling the exposure variable $O_{ij}$ by the Bernoulli exposure model.

## 2.3 Prior Predictive Matching (PPM)

Prior Predictive Matching (PPM) is a technique in Bayesian statistics that aims to specify prior distributions that incorporate the predictive features of observed data. The primary objective of PPM is to ensure that the selected prior information aligns with the patterns anticipated in the data. PPM operates by utilizing the prior predictive distribution to estimate certain meaningful virtual statistics that are inferred from the prior. Subsequently, it learns the optimal hyperparameters by aligning virtual statistics with the target value. In essence, PPM involves an iterative process where the prior distributions are adjusted based on the behavior of observed data to refine the Bayesian model for coherence and informativeness. Researchers apply the PPM method to probabilistic models, which can lead to more accurate inferences and predictions. In the context of PMF models, PPM helps in selecting prior hyperparameters that directly reflect the distributions of the observed data. For instance, da Silva et al. (2023) introduced PPM to set hyperparameters in PMF models, ensuring that the priors are well-grounded in empirical observations. This approach not only enhances the robustness of the model but also improves the interpretability of the results by aligning the prior beliefs with limited observed data.

## 3 Proposed Method

Most of the PMF models randomly initialize the prior distributions' hyperparameters and repeatedly search for the optimal values. The quality of posterior inference can be significantly influenced by the choice of the prior distribution and its hyperparameters. While da Silva et al. (2023) proposed a prior specification method using Prior Predictive Matching (PPM) to estimate the hyperparameters instead of grid search, we observed that the relative error of estimation increases substantially as the data becomes sparser. This is because the statistics observed from sparse datasets are inaccurate without taking into account the data sparsity in the model. Inspired by the concept of exposure introduced by Liang et al. (2016), we integrate the generic PMF with an additional exposure component to simulate the scenario of missing data. The model structure we propose, in contrast to the established exposure-based model detailed in Equation (4), is more generalized and accommodates varying non-zero prior hyperparameters and non-Gaussian observation distributions. This generic model framework is beneficial for our further analysis and inference of the estimator in closed-form with the PPM method. Consequently, our proposed estimator derived from the exposure-based model is more flexible to handle sparse data and achieves a lower relative error in experiments.

## 3.1 Model Definition

We define an exposure-based generic Bayesian MF model that incorporates data sparsity to derive improved prior specification estimators:

$$p_{ik} \sim F(\mu_p, \sigma_p^2), \quad q_{jk} \sim F(\mu_q, \sigma_q^2),$$
$$R_{ij} \sim F_R \left( \sum_{k=1}^K p_{ik} q_{jk}, \sigma^2 \right), \quad \text{with } \mathbb{E}[R_{ij}] = \sum_{k=1}^K p_{ik} q_{jk}. \tag{5}$$

Then we define an exposure random variable $O_{ij}$ and the corresponding observation as:

$$O_{ij} \sim F_{Expo}, \quad Y_{ij} = O_{ij}R_{ij}. \tag{6}$$

The $O_{ij}$ follows the exposure distribution $F_{Expo}$. In this research, we use the Bernoulli distribution if there is no extra explanation, $O_{ij} \sim Bern(\theta_{ij})$. The Figure 6 shows the complete structure of the model.

The relationship between variable $Y_{ij}$ and $R_{ij}$:

$$\begin{cases} Y_{ij}|O_{ij} = 1 \sim F_r \left( \sum_{k=1}^{K} p_{ik}q_{jk}, \sigma^2 \right) & ; \quad Y_{ij} = R_{ij}, \\ Y_{ij}|O_{ij} = 0 \sim \delta_0 & ; \quad P(Y_{ij} = 0|O_{ij} = 0) = 1. \end{cases} \tag{7}$$

### 3.2 Common Assumptions

We describe common assumptions which will be integrated in the specific model as combinations.

**Assumption 1** (A1). *Referring to Equation (6), assume the exposure variable $O_{ij}$ is independent of the true relevance $R_{ij}$, then we have:*

$$\mathbb{E}[Y_{ij}] = \mathbb{E}[O_{ij}R_{ij}] = \theta_{ij}\mathbb{E}[R_{ij}]$$
$$\longrightarrow \mathbb{E}[R_{ij}] = \theta_{ij}^{-1}\mathbb{E}[Y_{ij}]; \tag{8}$$
$$\mathbb{V}[Y_{ij}] = \mathbb{V}[O_{ij}R_{ij}] = \mathbb{E}[O_{ij}]^2\mathbb{V}[R_{ij}] + \mathbb{E}[R_{ij}]^2\mathbb{V}[O_{ij}] + \mathbb{V}[O_{ij}]\mathbb{V}[R_{ij}]$$
$$\longrightarrow \mathbb{V}[R_{ij}] = \theta_{ij}^{-1}\mathbb{V}[Y_{ij}] + \theta_{ij}^{-2}(\theta_{ij} - 1)\mathbb{E}[Y_{ij}]^2. \tag{9}$$

This assumption ensures that the exposure process does not influence the expected value of $R_{ij}$ and allows for straightforward factorization in expectation and variance calculations. The independence assumption simplifies statistical analysis by ensuring that $O_{ij}$ does not introduce additional variability or structure into $R_{ij}$.

**Assumption 2** (A2). *With reference to Equation (5), assume the observation model $F_R$ is Poisson distribution, then we have:*

$$\mathbb{E}[\mathbb{V}[R_{ij}|P, Q]] = \mathbb{E}[R_{ij}]. \tag{10}$$

The Poisson assumption is commonly used in count-based models, where the number of events follows a discrete stochastic process. This is natural in situations where the relevance score is derived from user interactions, such as clicks or purchase counts, which often exhibit Poisson-like behavior, especially when event occurrences are independent and event rates are approximately constant.

This assumption leverages the key property of Poisson distributions, where the variance equals the mean. By assuming a Poisson observation model, the expected conditional variance of $R_{ij}$ given $P$ and $Q$ is directly linked to its mean. This facilitates inference and simplifies the characterization of uncertainty in $R_{ij}$.

We also make two assumptions about the relationship between different exposure random variables.

**Assumption 3** (A3). *Assume that random variable $O_{ij}$ and $O_{tl}$ follows exactly the same distribution $O_{ij}, O_{tl} \sim Bern(\theta)$, then we have:*

$$\mathrm{Cov}[O_{ij}, O_{tl}] = \mathrm{Var}(\theta) = \theta(1 - \theta). \tag{11}$$

This assumption introduces dependence between exposure variables, meaning that observing one exposure event influence each other. In practical situations, it's a natural assumption that exposure variables are correlated with each other, depending on underlying latent factors. For instance, if one item has a high exposure probability, the others should have lower probabilities because of the limited total number of allowed exposure events, such as the number of items displayed on a single web page.

**Assumption 4** (A4). *Assume that random variable $O_{ij}$ and $O_{tl}$ are independent and identically distributed, $O_{ij}, O_{tl} \overset{iid}{\sim} Bern(\theta)$, then we have:*

$$\text{Cov}[O_{ij}, O_{tl}] = 0. \tag{12}$$

This assumption introduces independence between exposure variables, meaning that observing one exposure event does not influence another. This is a common assumption in probabilistic modeling when each observation is independently drawn. The independence simplifies variance calculations and removes potential correlations in the exposure process, making statistical inference more tractable.

### 3.3 Model Interpretation

**True relevance** Our proposed model differs from the original ExpoMF discussed in Section 2.2. We introduce an intermediate variable, $R_{ij}$, to distinguish it from the variable $Y_{ij}$. Saito et al. (2020) interprets $Y_{ij}$ as observations and $R_{ij}$ as true relevance, which may reflect actual preferences. Therefore, high-quality recommendations should be based on the true relevance $R_{ij}$ rather than solely observations.

**Data sparsity simulation** Our proposed model incorporates the idea of data exposure to simulate the data sparsity issues in synthetic datasets. By strategically adjusting the level of exposure, we can create various scenarios that simulate real-world conditions where data may be limited or unbalancedly distributed. This approach enables us to rigorously test the performance of estimators in a wider range of sparsity levels. In real-world scenarios, the reasons for missing data are intricate and unpredictable. For instance, users might not provide exposed item ratings because of their personality. While the concept of sparsity simulation is inspired by the exposure-based model, we hypothesize that it possesses remarkable capabilities to handle complex situations beyond the limitations of exposure alone.

**Exposure model selection** In the Exposure MF framework Liang et al. (2016), the absence of user ratings doesn't necessarily indicate a user's disinterest in a particular item. The likelihood of a rating being missing is significantly influenced by the extent to which the item has been exposed to the user. Liang et al. (2016) proposed two types of exposure models that determine the probability of item exposure. The first model is based on item popularity, suggesting that popular items are more likely to be exposed to users. The second model incorporates additional information, such as text topics or user locations, as exposure invariants. In this research, we treat the exposure variable as a Bernoulli variable. In real-world scenarios, we can replace the exposure variable with specific exposure models.

### 3.4 Generic Propositions

In our proposed model, we concentrate on the actual relevance $R_{ij}$ rather than the observed values $Y_{ij}$. We begin by deriving fundamental principles in a general form, which we later use to derive practical results with specific assumptions. The derivation relies on the model definition provided by Equation (5) and the fundamental laws of total expectation, variance, and covariance.

To begin, we establish key statistical properties of the relevance values $R_{ij}$, including their expected value and variance. This serves as the foundation for understanding how the structure of $R_{ij}$ influences the relationships between different entries.

**Proposition 1.** *For any entry of the true relevance matrix $R_{ij} \in \mathbb{R}^{I \times J}$, the mean and variance in generic form is given as follows:*

$$\mathbb{E}[R_{ij}] = K\mu_p\mu_q, \tag{13}$$

$$\mathbb{V}[R_{ij}] = K[(\mu_p\sigma_q)^2 + (\mu_q\sigma_p)^2 + (\sigma_p\sigma_q)^2] + \mathbb{E}[\mathbb{V}[R_{ij}|P,Q]]. \tag{14}$$

These fundamental statistical properties provide insight into how individual entries of $R_{ij}$ behave under the given model assumptions. However, in many applications, understanding the correlation structure between

different entries is also crucial. We now analyze the prior predictive correlation between pairs of entries. To be clarified, the following $\rho$ denotes the original correlation coefficients sampled from the dataset. The $\rho^*$ denotes the modified $\rho$ in our different models. The $\rho_1^*$ and $\rho_2^*$ denote the raw/column $\rho^*$ respectively.

**Proposition 2.** *For any pair of entries $R_{ij}$ and $R_{tl}$, the prior predictive correlation is given as follows:*

$$\rho[R_{ij}, R_{tl}] = \begin{cases} 0, & \text{if} \quad i \neq t \quad \& \quad j \neq l; \\ 1, & \text{if} \quad i = t \quad \& \quad j = l; \\ \rho_1^* = \frac{K(\mu_q \sigma_p)^2}{\mathbb{V}[R_{ij}]}, & \text{if} \quad i = t \quad \& \quad j \neq l; \\ \rho_2^* = \frac{K(\mu_p \sigma_q)^2}{\mathbb{V}[R_{ij}]}, & \text{if} \quad i \neq t \quad \& \quad j = l. \end{cases} \tag{15}$$

This result characterizes how the dependencies between different entries of $R_{ij}$ are structured within the model. In particular, it shows that direct dependencies exist when either the row index or the column index is shared, with specific correlation coefficients.

Given this structure, we now extend our analysis to the observed values $Y_{ij}$, which depend not only on $R_{ij}$ but also on the observation process itself. We derive the covariance structure of $Y_{ij}$ in the following proposition.

**Proposition 3.** *For any pair of entries $Y_{ij}$ and $Y_{tl}$, the generic expansion of the two entries is given as follows:*

$$\begin{aligned} \text{Cov}[Y_{ij}, Y_{tl}] &= \text{Cov}[O_{ij} R_{ij}, O_{tl} R_{tl}] \\ &= \text{Cov}[O_{ij}, O_{tl}] \text{Cov}[R_{ij}, R_{tl}] + \mathbb{E}[R_{ij}] \mathbb{E}[R_{tl}] \text{Cov}[O_{ij}, O_{tl}] + \mathbb{E}[O_{ij}] \mathbb{E}[O_{tl}] \text{Cov}[R_{ij}, R_{tl}]. \end{aligned} \tag{16}$$

This result shows how the covariance of the observed values $Y_{ij}$ is influenced by both the underlying relevance structure and the observation process.

### 3.5 Particular Models

We describe two particular models to derive practical results. Refer to Appendix A for proof details. In this stage, we focus on the estimator of the number of latent factors.

**Model 1** (M1)**.** *With generic model defined in Section 3.1 and A1, A2 A3, we can solve that:*

$$\rho^* = \frac{\mathbb{V}[Y_{ij}] \rho - \theta(1 - \theta) \mathbb{E}[R_{ij}]^2}{\theta \mathbb{V}[R_{ij}]}. \tag{17}$$

*And the new estimator $\hat{K}$ is written as:*

$$\hat{K} = \frac{(1 - (\rho_1^* + \rho_2^*)) \mathbb{V}[R_{ij}] - \mathbb{E}[R_{ij}]}{\rho_1^* \rho_2^*} \left( \frac{\mathbb{E}[R_{ij}]}{\mathbb{V}[R_{ij}]} \right)^2. \tag{18}$$

**Model 2** (M2)**.** *With generic model defined in Section 3.1 and A1, A2 A4, we can solve that:*

$$\rho^* = \theta^{-2} \rho \tag{19}$$

*And the new estimator $\hat{K}$ is written as:*

$$\hat{K} = \frac{(1 - (\rho_1^* + \rho_2^*)) \mathbb{V}[R_{ij}] - \mathbb{E}[R_{ij}]}{\rho_1^* \rho_2^*} \left( \frac{\mathbb{E}[R_{ij}]}{\mathbb{V}[R_{ij}]} \right)^2. \tag{20}$$

The estimators of $\hat{K}$ can be represented by five statistics including $\mathbb{E}[Y_{ij}]$, $\mathbb{V}[Y_{ij}]$, the row/column correlation coefficients $\rho_1 / \rho_2$ and parameter $\theta$ which sampled from the observation matrix $Y$.

## 4    Experiments

In this section, we reproduce the experiment conducted in the original work (da Silva et al., 2023) in advance. To conduct a practical experiment, we need to determine the prior distribution beforehand. For simplicity and demonstration purposes, we use the probability mass function (PMF) for two reasons: (1) The gamma prior is conjugate to the Poisson likelihood; (2) The Poisson distribution has the same mean and variance (described in Equation (46)).

### 4.1    Hyperparameters specification

Referring to the original work, we consider initializing the hyperparameters with the following specifications ($a/c$ are the *shape* parameters and $b/d$ are the *rate* parameters of Gamma distributions):

Table 1: Hyperparameters Initialization for Different Specifications

| Spec. | $a$ | $b$ | $c$ | $d$ | $\mu_p$ | $\sigma_p$ | $\mu_q$ | $\sigma_q$ | $\mathbb{E}[R]$ | $\mathbb{V}[R]$ |
|---|---|---|---|---|---|---|---|---|---|---|
| A | 10 | 1 | 10 | 1 | 10.0 | 3.16 | 10.0 | 3.16 | 2500.00 | 55000.00 |
| D | 0.1 | 1 | 0.1 | 1 | 0.1 | 0.32 | 0.1 | 0.32 | 0.25 | 0.55 |
| F | 1 | 1 | 0.1 | 0.1 | 1.0 | 1.0 | 1.0 | 3.16 | 25.00 | 550.00 |

We conduct the experiments with specifications A, D, and F because they are distinct from each other. The full specification setup defined by da Silva et al. (2023) is described in Table 4. In specification A, matrices $P$ and $Q$ share the same prior parameters, but their shape parameters are 10 times larger than their rate parameters. In specification D, matrices $P$ and $Q$ also share the same prior parameters, but their shape parameters are 10 times smaller than their rate parameters. In specification F, the shape parameters are the same for both distributions, but the prior parameters of matrix $P$ are numerically 10 times larger than those of matrix $Q$.

### 4.2    Experimental Steps

We first generate the synthetic data with the following 3 steps repeatedly: (1) We sample the matrix $P$ and $Q$ with the prior hyperparameters for particular specifications; (2) We recover the fully dense matrix $R$ by the product of $P^\top$ and $Q$; (3) We sample the Bernoulli variables $O_{ij}$ with different sparsity levels and multiply them with each entry of the dense matrix $R$ to obtain the sparse observation matrix $Y$.

Table 2 outlines the three variables we modify to set up our experiments. We employ prior specifications A, D, and F, as outlined in Table 1. We alter the true value of $K$, representing the number of latent factors, from 25 to 150. Additionally, we change the parameter Pobs of the Bernoulli distribution into three magnitude groups. Group 1, the only setup used in the original work, produces excessively dense observations. To enhance the realism of our simulations, we introduce Groups 2 and 3, which generate more sparse observation matrices $Y$. Consequently, we generate 30 records for each setup, resulting in a total of 8,640 data records.

Table 2: Variables of Experiment Setups

| Prior Spec. | K (Num. of Latent Factors) | Pobs. (Parameter of Bernoulli distribution) |
|---|---|---|
| [A, D, F] | [25, 50, 75, 100, 125, 150] | Group 1: [1.0, 0.98, 0.96, 0.94, 0.92, 0.90]
Group 2: [0.5, 0.4, 0.3, 0.2, 0.1]
Group 3: [0.05, 0.04, 0.03, 0.02, 0.01] |

Then we continue our experiment with the following steps: (4) We sample the statistics from the observation matrix $Y$ including $\mathbb{E}[Y_{ij}]$, $\mathbb{V}[Y_{ij}]$ and the row/column correlation coefficients $\rho_1/\rho_2$; (5) We calculate the

estimations of $K$ ($\hat{K}$) with different models; (6) We measure the model performance by the relative error between the true $K$ and the estimated $\hat{K}$ ($\frac{K-\hat{K}}{K}\%$).

## 4.3 Results

The figures presented in this section, Figures 1, 2, and 3, illustrate the outcomes of the experiments conducted using the previously proposed method. These figures are quantile plots that represent the relative error between $\hat{K}$ and $K$ for each model, with *Pobs* ranging from 0.01 to 1.0. The cross-over points indicate the median values. Due to the large and dispersed results with a 95% percentile, we plot with a 50% percentile as error bars for better visualization. Table 3 presents the median values for each specification, and the lowest absolute values among model comparisons are highlighted. We will state our observations and discuss the results in the next section.

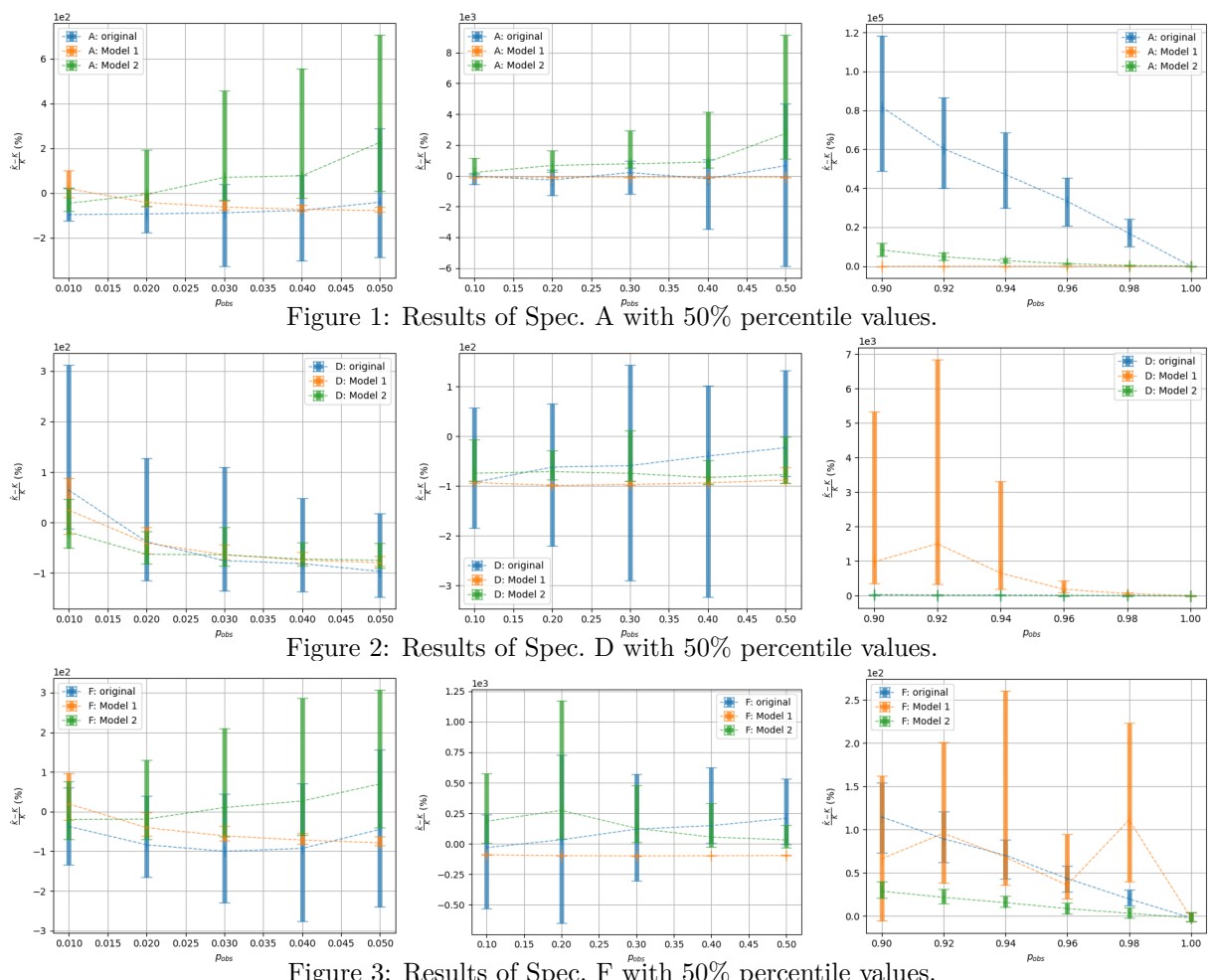

Figure 1: Results of Spec. A with 50% percentile values.

Figure 2: Results of Spec. D with 50% percentile values.

Figure 3: Results of Spec. F with 50% percentile values.

Table 3: The Median Value of Relative Errors for All Experiments

| | | | | | | | | | *Pobs* | | | | | | | | | |
|---|---|---|---|---|---|---|---|---|---|---|---|---|---|---|---|---|---|---|
| | | | 0.01 | 0.02 | 0.03 | 0.04 | 0.05 | 0.1 | 0.2 | 0.3 | 0.4 | 0.5 | 0.90 | 0.92 | 0.94 | 0.96 | 0.98 | 1.00 |
| | | *Original* | -96.17 | -93.82 | -88.12 | -77.65 | -41.00 | -42.40 | -277.16 | 202.87 | -203.04 | 649.61 | 81709.87 | 60304.04 | 47040.06 | 33297.74 | 16836.48 | -0.16 |
| | *Spec. A* | *Model 1* | 19.46 | -42.00 | -62.61 | -73.08 | -78.72 | -94.12 | -98.09 | -99.65 | -99.46 | -97.68 | -77.44 | -71.26 | -60.83 | -39.29 | 30.59 | -0.16 |
| *Median* | | *Model 2* | -45.89 | -6.15 | 70.07 | 77.29 | 226.53 | 209.70 | 665.91 | 772.65 | 883.01 | 2740.34 | 8007.24 | 4773.18 | 2813.04 | 1242.95 | 142.52 | -0.16 |
| *Value* | | *Original* | 64.01 | -37.90 | -75.35 | -81.05 | -96.62 | -92.72 | -62.09 | -59.07 | -39.79 | -23.04 | 19.25 | 12.82 | 9.74 | 4.37 | -0.51 | -4.42 |
| *of* | *Spec. D* | *Model 1* | 24.55 | -40.31 | -63.69 | -73.86 | -79.41 | -92.77 | -98.92 | -96.93 | -93.66 | -88.22 | 775.32 | 1264.55 | 565.97 | 160.80 | 50.89 | -4.42 |
| *Relative* | | *Model 2* | -18.63 | -62.81 | -63.76 | -72.26 | -74.53 | -74.32 | -71.01 | -74.64 | -82.95 | -76.99 | -3.33 | -5.49 | -3.09 | -4.88 | -5.25 | -4.42 |
| *Error (%)* | | *Original* | -37.42 | -83.81 | -100.07 | -92.94 | -45.08 | -31.95 | 32.39 | 120.00 | 148.33 | 208.37 | 114.74 | 89.42 | 70.38 | 43.50 | 19.82 | -1.90 |
| | *Spec. F* | *Model 1* | 19.39 | -40.62 | -61.24 | -71.87 | -78.89 | -91.09 | -97.32 | -99.78 | -98.30 | -96.31 | 40.29 | 69.58 | 53.50 | 27.50 | 103.70 | -1.90 |
| | | *Model 2* | -20.07 | -19.09 | 10.19 | 26.64 | 69.53 | 184.43 | 273.67 | 126.84 | 54.95 | 31.30 | -13.01 | -8.97 | -4.17 | -2.61 | -1.97 | -1.90 |

## 5   Discussions

### 5.1   Observations of Experiment Results

We evaluate the ratio between our estimated $\hat{K}$ and the true $K$:

$$\frac{\hat{K}}{K} = \frac{\rho_1\rho_2}{\rho_1^*\rho_2^*} \left(\frac{\mathbb{E}[R_{ij}]\mathbb{V}[Y_{ij}]}{\mathbb{E}[Y_{ij}]\mathbb{V}[R_{ij}]}\right)^2 \frac{(1-(\rho_1^*+\rho_2^*))\mathbb{V}[R_{ij}]-\mathbb{E}[R_{ij}]}{(1-(\rho_1+\rho_2))\mathbb{V}[Y_{ij}]-\mathbb{E}[Y_{ij}]}. \tag{21}$$

Then we separately analyze the following term for the different models.

**Model 1**   We can write $\rho^*$ as:

$$\rho^* = \frac{\rho + (1-\theta^{-1})(\frac{\mathbb{E}[Y]^2}{\mathbb{V}[Y]})}{1 + (1-\theta^{-1})(\frac{\mathbb{E}[Y]^2}{\mathbb{V}[Y]})}. \tag{22}$$

We denote the denominator $1 + (1-\theta^{-1})(\frac{\mathbb{E}[Y]^2}{\mathbb{V}[Y]})$ as $\tau$ for simplicity, then Equation (21) can be rewritten as:

$$\frac{\hat{K}}{K} = \frac{\rho_1\rho_2}{(\rho_1-1+\tau)(\rho_2-1+\tau)} \frac{(2-\tau-(\rho_1+\rho_2))-\frac{\mathbb{E}[Y]}{\mathbb{V}[Y]}}{(1-(\rho_1+\rho_2))-\frac{\mathbb{E}[Y]}{\mathbb{V}[Y]}}. \tag{23}$$

There are restrictions in the above equations to avoid zero denominators:

1. In the formula of $\rho^*$, the root of $\tau = 0$ always exists. For simplicity of analysis, we assume the term $\frac{\mathbb{E}[Y]}{\mathbb{V}[Y]}$ as constant once it is sampled. The value of $\tau$ is inversely proportional to the value of $\theta$ with upper bound of 1.

2. In Equation (23), the denominators $(\rho_1-1+\tau)$ and $(\rho_2-1+\tau)$ are possible to be zero. Because the two terms have equal forms, we use $\rho_1$ for discuss. The existence of the root of equation $(\rho_1-1+\tau)$ depends on the sampled value of $\rho_1$. We assume the root exists, then the closed form can be written as:

$$\theta_0 = \frac{1}{1 + (\frac{\mathbb{V}[Y]}{\mathbb{E}[Y]^2})\rho_1}. \tag{24}$$

We use the true expectations and variances for visualization. Because the sampled value of $\rho_1$ is difficult to predict, we calculate the $\theta_0$ according to the theoretical range $[-1, 1]$ of the correlation coefficient.

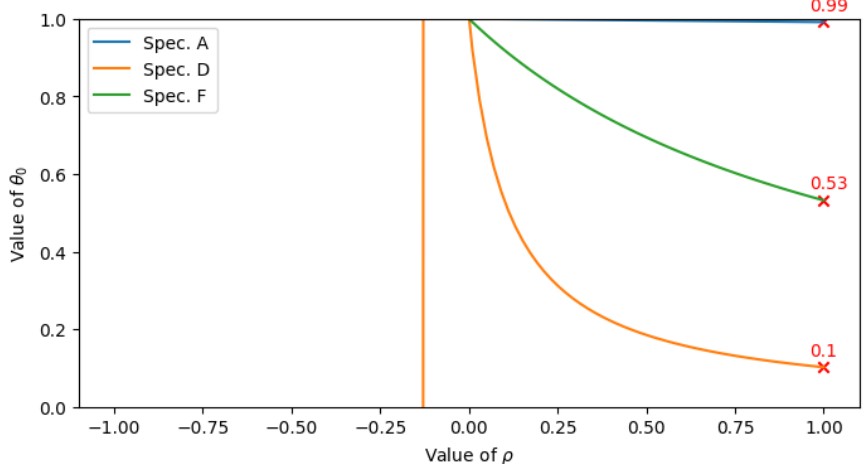

Figure 4: The relationship between $\rho$ and $\theta_0$ with different specifications

The Figure 4 shows the variation of the root $\theta_0$ corresponding to different $\rho$ values. Because the valid range of $\theta_0$ is $[0,1]$, we discard the invalid values in the graphs. For Spec. A, the value of $\theta_0$ remains around 1 for all $\rho$ values. This indicates that Model 1 performs best in Spec. A because the minimal value of its $\theta_0$ is larger than 0.99. However, for any valid value of $\theta_0$, there exists at least one corresponding $\rho$ causing a zero denominator in Spec. D. For Spec. F, when $\theta_0$ is smaller than 0.53 in Spec.5, both of the the denominators $(\rho_1 - 1 + \tau)$ and $(\rho_2 - 1 + \tau)$ are impossible to be zero. When $\theta_0 > 0.53$, Model 1 performs better in Spec. F than Spec. D because there are two corresponding $\rho$ values causing a zero denominator.

3. In Equation (23), the last denominator $(1 - (\rho_1 + \rho_2)) - \frac{\mathbb{E}[Y]}{\mathbb{V}[Y]}$ is possible to be zero, which is independent of the variable $\theta$.

Refer to the terms in Equation (23), the primary factors affecting the relative error are the variable $\tau$ and particular scenarios that result in a zero denominator. Essentially, the ratio increases as $\theta$ decreases until a zero denominator arises. After this, the ratio falls to the limitation as $\theta$ approaches zero, indicating that the $\hat{k}$ also reaches to the the lowest valid value for estimation.

$$\lim_{\theta \to 0} \frac{\hat{K}}{K} = \frac{\rho_1 \rho_2}{(\rho_1 + \frac{\mathbb{E}[Y]^2}{\mathbb{V}[Y]})(\rho_2 + \frac{\mathbb{E}[Y]^2}{\mathbb{V}[Y]})} \left( 1 - \frac{\frac{\mathbb{E}[Y]^2}{\mathbb{V}[Y]}}{(1 - (\rho_1 + \rho_2)) - \frac{\mathbb{E}[Y]}{\mathbb{V}[Y]}} \right). \tag{25}$$

**Model 2** We can write $\rho^*$ as:

$$\rho^* = \frac{\theta^{-1}\rho}{1 + (1 - \theta^{-1})(\frac{\mathbb{E}[Y]^2}{\mathbb{V}[Y]})}. \tag{26}$$

We denote the denominator $1 + (1 - \theta^{-1})(\frac{\mathbb{E}[Y]^2}{\mathbb{V}[Y]})$ as $\tau$ for simplicity, then Equation (21) can be rewritten as:

$$\frac{\hat{K}}{K} = \frac{(\tau - \theta^{-1}(\rho_1 + \rho_2)) - \frac{\mathbb{E}[Y]}{\mathbb{V}[Y]}}{(1 - (\rho_1 + \rho_2)) - \frac{\mathbb{E}[Y]}{\mathbb{V}[Y]}} \theta. \tag{27}$$

Different from the ratio form of Model 1, Equation (27) only has one denominator independent of variable $\theta$, which means Model 2 is less likely to have a zero denominator.

## 5.2 Analysis of Correlation Coefficients

The Pearson correlation coefficient, employed in the derivation, ranges from -1 to +1. In the matrix factorization problem, we calculate the correlation coefficients between two entries, as specified in Equation (2), under four distinct scenarios. Consequently, we obtain non-negative values for $\rho_1$ and $\rho_2$ in accordance with the general matrix factorization model presented in Equation (5). In the intermediate proof showed in Equation (42), the covariance when $i = t$ is that: $\text{Cov}[p_{ik}q_{jk}, p_{tk}q_{lk}] = \mu_q^2 \sigma_p^2$. Because the generic model assumes that the mean values $\mu_p$ and $\mu_q$ are constant, the equation $\mathbb{E}[q_{jk}] = \mathbb{E}[q_{lk}]$ implies that the covariance matrix is non-negative. However, in practice, it is possible to obtain negative sampled correlation coefficients from sparse datasets. In real-world applications, negative correlation coefficients are more realistic. For example, in user-item recommendation scenarios, we interpret $\rho_1$ as the correlation between rating scores from different items for the same user. This conflict between the assumption of constant mean values and the possibility of negative correlation coefficients can lead to ambiguous estimation of hyperparameters, such as a negative value for $K$.

This section highlights a limitation of the generic matrix factorization model: it lacks the ability to capture the inverse relationship between latent features. Naturally, we set a hyper-prior for the hyperparameters $\mu_p$ and $\mu_q$ instead of a constant. For the random variable $q_{jk} \sim F(\mu_q, \sigma_q^2)$, where $F$ is an unknown distribution and $\mu_q \sim G(\lambda_q, \tau^2)$, where $G$ is also an unknown distribution, we assume that both distributions are unknown.

We denote the resulting distribution as $H$, which is a compound distribution of $F$ and $G$. To integrate out the unknown parameter $\lambda$, we write the probability density function as follows:

$$p_H(q_{jk}) = \int p_F(q_{jk}|\lambda_q)p_G(\lambda_q)dq. \tag{28}$$

According to the law of total expectation:

$$\mathbb{E}_H[q_{jk}] = \mathbb{E}_G[\mathbb{E}_F[q_{jk}|\mu_q]]. \tag{29}$$

If $\mu_q$, the mean of $F$ is distributed as $\lambda_q$, the mean of $G$, the above equation informs that the unconditional expectation $\mathbb{E}_H[q_{jk}] = \lambda_q$, and $\mathbb{E}_H[q_{lk}] = \lambda_q$. However, the above covariance remains non-negative in quadratic form equal to $\lambda_q^2\sigma_p^2$.

### 5.3 Correlated Latent Features

To overcome the limitation of non-negative correlation coefficients in the generic models, we propose an essential solution with the following new assumption: *for any valid combinations indexed by $i, j, t, l$, any two random variables from the same latent feature vector are dependent. Otherwise, they are independent.* Assume latent factors follow Multivariate Distribution denoted as $\mathcal{F}_k$:

$$p_i \sim \mathcal{F}_k(\mu_p, \Sigma_p), \quad q_j \sim \mathcal{F}_k(\mu_q, \Sigma_q), \tag{30}$$

finally, we obtain the closed form of the covariance,

$$\text{Cov}[R_{ij}, R_{il}] = \mu_q^2(\mathbf{1}^T\Sigma_p\mathbf{1}).$$

The above notation refers to the all-ones vector. However, the covariance matrix is always positive semi-definite, and the covariance of any two entries is always non-negative. Details of derivation are described in A.7.

## 6 Conclusion

In conclusion, the exploration of Bayesian Matrix Factorization (MF) models highlights the versatility and adaptability of the framework in addressing various data-related challenges, particularly in the context of missing data. The distinct models presented, including conventional Probabilistic MF, Poisson MF, and Bayesian Probabilistic MF, illustrate how different prior choices and observation models can effectively capture the latent patterns and uncertainties inherent in user-item interactions.

The introduction of the Exposure-based Model (ExpoMF) further enriches this discussion by addressing the complexities of data sparsity and the positive-unlabeled problem, providing a nuanced understanding of implicit feedback mechanisms. Adopting an exposure variable allows for a more accurate representation of user preferences and item visibility, enhancing recommendation systems.

The Prior Predictive Matching (PPM) technique serves as a critical tool for ensuring that the prior distributions align with observed data patterns, ultimately leading to more robust and interpretable models. This iterative approach emphasizes the importance of grounding Bayesian models in empirical evidence, paving the way for improved predictive performance and insights. Collectively, these advancements contribute to a deeper understanding of Bayesian MF and its applications in modern recommendation systems, setting a solid foundation for future research and development in this field.

Overall, these advancements not only enrich our comprehension of Bayesian MF but also lay a strong groundwork for ongoing innovation within recommendation system research, fostering future developments that could further refine the intersection of data analysis and user experience.

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

# A Proof

## A.1 Preliminaries

In this section, we introduce the preliminaries for proof including the Law of Total Expectation, the Law of Total Variance, and the Law of Total Covariance. For two random variable $X$ and $Y$, we know that:

$$\mathbb{E}[Y] = \mathbb{E}[\mathbb{E}[Y|X]] \tag{31}$$

$$\mathbb{V}[Y] = \mathbb{E}[\mathbb{V}[Y|X]] + \mathbb{V}[\mathbb{E}[Y|X]] \tag{32}$$

$$Cov[X,Y] = \mathbb{E}[Cov[X,Y]|Z] + Cov[\mathbb{E}[X|Z], \mathbb{E}[Y|Z]] \tag{33}$$

$$\mathbb{E}[X^2] = \mathbb{V}[X] + \mathbb{E}[X]^2 \tag{34}$$

$$\mathbb{V}\left[\sum_k X_k\right] = \sum_k \mathbb{V}[X_k] + 2\sum_{k<k'} Cov[X_k, X_{k'}]. \tag{35}$$

If $X$ and $Y$ are independent:

$$\mathbb{V}[XY] = \mathbb{E}[X^2]\mathbb{E}[Y^2] - \mathbb{E}[X]^2\mathbb{E}[Y]^2 \tag{36}$$

$$\mathbb{V}[XY] = \mathbb{E}[X]^2\mathbb{V}[Y] + \mathbb{E}[Y]^2\mathbb{V}[X] + \mathbb{V}[X]\mathbb{V}[Y]. \tag{37}$$

For four random variable $A$, $B$, $X$ and $Y$:

$$\text{Cov}[AX, BY] = \text{Cov}[A,B]\text{Cov}[X,Y] + \mathbb{E}[A]\mathbb{E}[B]\text{Cov}[X,Y] + \mathbb{E}[X]\mathbb{E}[Y]\text{Cov}[A,B]. \tag{38}$$

## A.2 Intermediate propositions

**Proposition 4.** *For any valid combinations indexed by $i, j, t, l$, if the latent indexes $k \neq k'$, then $Cov[p_{ik}q_{jk}, p_{tk'}q_{lk'}] = 0$.*

*Proof.* By definition of covariance:

$$\text{Cov}[p_{ik}q_{jk}, p_{tk'}q_{lk'}] = \mathbb{E}[p_{ik}q_{jk}p_{tk'}q_{lk'}] - \mathbb{E}[p_{ik}q_{jk}]\mathbb{E}[p_{tk'}q_{lk'}].$$

Because when $k \neq k'$, $\mathbb{E}[p_{ik}q_{jk}p_{tk'}q_{lk'}] = \mathbb{E}[p_{ik}q_{jk}]\mathbb{E}[p_{tk'}q_{lk'}]$, then we have $Cov[p_{ik}q_{jk}, p_{tk'}q_{lk'}] = 0$. □

**Proposition 5.** *For any valid combinations indexed by $i, j, t, l$, the following equations holds:*

$$\mathbb{E}[\sum_k p_{ik}q_{jk}] = K\mu_p\mu_q \tag{39}$$

$$\mathbb{V}[\sum_k p_{ik}q_{jk}] = K[(\mu_p\sigma_q)^2 + (\mu_q\sigma_p)^2 + (\sigma_p\sigma_q)^2] \tag{40}$$

$$\text{Cov}\left[\sum_k p_{ik}q_{jk}, \sum_k p_{tk}q_{lk}\right] = K[\delta_{it}(\mu_p\sigma_q)^2 + \delta_{jl}(\mu_q\sigma_p)^2 + \delta_{it}\delta_{jl}(\sigma_p\sigma_q)^2]. \tag{41}$$

*Proof.*

- Because in our model, $p_{ik}$ and $q_{jk}$ are independent, then $\mathbb{E}[\sum_k p_{ik}q_{jk}] = \sum_k \mathbb{E}[p_{ik}]\mathbb{E}[q_{jk}] = K\mu_p\mu_q$;

- Referring to Equation (35), we know that:

$$\mathbb{V}\left[\sum_k p_{ik}q_{jk}\right] = \sum_k \mathbb{V}[p_{ik}q_{jk}] + 2\sum_{k<k'} \text{Cov}[p_{ik}q_{jk}, p_{tk'}q_{lk'}].$$

And referring to Proposition 4, we know that the second term of above equation will be zero. Then we have:

$$\mathbb{V}\left[\sum_k p_{ik}q_{jk}\right] = \sum_k \mathbb{V}[p_{ik}q_{jk}].$$

Because $p_{ik}$ and $q_{jk}$ are independent, recall the Equation (37), we know that:

$$\mathbb{V}[p_{ik}q_{jk}] = \mathbb{E}[p_{ik}]^2\mathbb{V}[q_{jk}] + \mathbb{E}[q_{jk}]^2\mathbb{V}[p_{ik}] + \mathbb{V}[p_{ik}]\mathbb{V}[q_{jk}]$$
$$= (\mu_p\sigma_q)^2 + (\mu_q\sigma_p)^2 + (\sigma_p\sigma_q)^2.$$

Thus we obtain:

$$\mathbb{V}\left[\sum_k p_{ik}q_{jk}\right] = K[(\mu_p\sigma_q)^2 + (\mu_q\sigma_p)^2 + (\sigma_p\sigma_q)^2];$$

- Recall the definition of covariance:

$$\text{Cov}\left[\sum_k p_{ik}q_{jk}, \sum_k p_{tk}q_{lk}\right] = \mathbb{E}\left[\sum_{k,k'} p_{ik}q_{jk}p_{tk'}q_{lk'}\right] - \mathbb{E}\left[\sum_k p_{ik}q_{jk}\right]\mathbb{E}\left[\sum_{k'} p_{tk'}q_{lk'}\right]$$
$$= \sum_{k,k'} \mathbb{E}[p_{ik}q_{jk}p_{tk'}q_{lk'}] - \mathbb{E}[p_{ik}]\mathbb{E}[q_{jk}]\mathbb{E}[p_{tk'}]\mathbb{E}[q_{lk'}]$$
$$= \sum_{k,k'} \text{Cov}[p_{ik}q_{jk}, p_{tk'}q_{lk'}].$$

Referring to Proposition 4, all the terms with different k are zero, thus:

$$\text{Cov}\left[\sum_k p_{ik}q_{jk}, \sum_k p_{tk}q_{lk}\right] = \sum_k \text{Cov}[p_{ik}q_{jk}, p_{tk}q_{lk}]. \tag{42}$$

Then for the same latent index $k$, there will be four different cases to calculate $Cov[p_{ik}q_{jk}, p_{tk}q_{lk}]$:

1. *if* $i \neq t$ & $j \neq l$: due to the independence of these variables, we know that

$$\text{Cov}[p_{ik}q_{jk}, p_{tk}q_{lk}] = 0;$$

2. *if* $i = t$ & $j \neq l$:

$$\text{Cov}[p_{ik}q_{jk}, p_{tk}q_{lk}] = \text{Cov}[p_{ik}q_{jk}, p_{ik}q_{lk}]$$
$$= \mathbb{E}[p_{ik}^2 q_{jk}q_{lk}] - \mathbb{E}[p_{ik}]^2\mathbb{E}[q_{jk}]\mathbb{E}[q_{lk}]$$
$$= \mathbb{E}[p_{ik}^2]\mathbb{E}[q_{jk}]\mathbb{E}[q_{lk}] - \mathbb{E}[p_{ik}]^2\mathbb{E}[q_{jk}]\mathbb{E}[q_{lk}]$$
$$= \mathbb{E}[q_{jk}]\mathbb{E}[q_{lk}](\mathbb{E}[p_{ik}^2] - \mathbb{E}[p_{ik}]^2)$$
$$= \mathbb{E}[q_{jk}]\mathbb{E}[q_{lk}]\mathbb{V}[p_{ik}]$$
$$= \mu_q^2\sigma_p^2;$$

3. *if* $i \neq t$ & $j = l$: similarly to above steps, we obtain

$$\text{Cov}[p_{ik}q_{jk}, p_{tk}q_{lk}] = \mu_p^2\sigma_q^2;$$

4. $if \quad i = t \quad \& \quad j = l$, referring to proof in last steps:

$$\begin{aligned}
\mathrm{Cov}[p_{ik}q_{jk}, p_{tk}q_{lk}] &= \mathrm{Cov}[p_{ik}q_{jk}, p_{ik}q_{jk}] \\
&= \mathbb{V}[p_{ik}q_{jk}] \\
&= (\mu_p\sigma_q)^2 + (\mu_q\sigma_p)^2 + (\sigma_p\sigma_q)^2.
\end{aligned}$$

For simplicity of presentation, the four cases can be written in one closed form with Kronecker delta (if $i = j$, $\delta_{ij} = 1$. Otherwise, $\delta_{ij} = 0$):

$$\mathrm{Cov}[p_{ik}q_{jk}, p_{tk}q_{lk}] = \delta_{it}(\mu_p\sigma_q)^2 + \delta_{jl}(\mu_q\sigma_p)^2 + \delta_{it}\delta_{jl}(\sigma_p\sigma_q)^2.$$

Finally, the last equation of this proposition holds:

$$\mathrm{Cov}\left[\sum_k p_{ik}q_{jk}, \sum_k p_{tk}q_{lk}\right] = K[\delta_{it}(\mu_p\sigma_q)^2 + \delta_{jl}(\mu_q\sigma_p)^2 + \delta_{it}\delta_{jl}(\sigma_p\sigma_q)^2].$$

$\square$

### A.3 Expected values and variance of true relevance

Now we calculate the prior predictive expected value and variance of true relevance $R_{ij}$. This part of model definition is described in Equation (5). And this part of proof corresponds to Proposition 1.

**Proposition 6.** *For any valid combinations indexed by $i, j, t, l$, the following equations holds:*

$$\mathbb{E}[R_{ij}] = K\mu_p\mu_q \tag{43}$$

$$\mathbb{V}[R_{ij}] = K[(\mu_p\sigma_q)^2 + (\mu_q\sigma_p)^2 + (\sigma_p\sigma_q)^2] + \mathbb{E}[\mathbb{V}[R_{ij}|P, Q]]. \tag{44}$$

*Specifically for the PMF model described in Equation (3), the variance of relevance can be simplified as:*

$$\mathbb{V}[R_{ij}] = K[(\mu_p\sigma_q)^2 + (\mu_q\sigma_p)^2 + (\sigma_p\sigma_q)^2 + \mu_p\mu_q]. \tag{45}$$

*Proof.*

- Because of the law of total expectation and results in Equation (39):

$$\begin{aligned}
\mathbb{E}[R_{ij}] &= \mathbb{E}\left[\mathbb{E}\left[R_{ij}\bigg| \sum_k p_{ik}q_{jk}\right]\right] \\
&= \mathbb{E}\left[\sum_k p_{ik}q_{jk}\right] = K\mu_p\mu_q;
\end{aligned}$$

- Because of the law of total variance (Equation (32)), we have:

$$\begin{aligned}
\mathbb{V}[R_{ij}] &= \mathbb{E}\left[\mathbb{V}\left[R_{ij}\bigg| \sum_k p_{ik}q_{jk}\right]\right] + \mathbb{V}\left[\mathbb{E}\left[R_{ij}\bigg| \sum_k p_{ik}q_{jk}\right]\right] \\
&= \mathbb{E}\left[\mathbb{V}\left[R_{ij}\bigg| \sum_k p_{ik}q_{jk}\right]\right] + \mathbb{V}\left[\sum_k p_{ik}q_{jk}\right].
\end{aligned}$$

Denote the latent parameters as $P, Q$, we finally obtain:

$$\mathbb{V}[R_{ij}] = K[(\mu_p\sigma_q)^2 + (\mu_q\sigma_p)^2 + (\sigma_p\sigma_q)^2] + \mathbb{E}[\mathbb{V}[R_{ij}|P, Q]];$$

- For experiments, we should determine the specific prior distribution. In the PMF model with Poisson distribution as observations, we have that:

$$\mathbb{E}\left[\mathbb{V}\left[R_{ij}\Big|\sum_k p_{ik}q_{jk}\right]\right] = \mathbb{E}\left[\mathbb{E}\left[R_{ij}\Big|\sum_k p_{ik}q_{jk}\right]\right] = \mathbb{E}\left[\sum_k p_{ik}q_{jk}\right] = \mu_p\mu_q. \tag{46}$$

Then we obtain the particular form for PMF:

$$\mathbb{V}[R_{ij}] = K[(\mu_p\sigma_q)^2 + (\mu_q\sigma_p)^2 + (\sigma_p\sigma_q)^2 + \mu_p\mu_q].$$

$\square$

## A.4 Covariance and correlation of true relevance

Now we calculate the covariance and correlation for the prior predictive distribution of true relevance $R_{ij}$. This part of the model definition is described in Equation (5). And this part of the proof corresponds to Proposition 2.

**Proposition 7.** *The prior predictive connivance of true relevance for the generic model is given by:*

$$\text{Cov}[R_{ij}, R_{tl}] = \delta_{it}\delta_{jl}\mathbb{E}[\mathbb{V}[R_{ij}|P,Q]] + K[\delta_{it}(\mu_p\sigma_q)^2 + \delta_{jl}(\mu_q\sigma_p)^2 + \delta_{it}\delta_{jl}(\sigma_p\sigma_q)^2]. \tag{47}$$

*In the PMF model with Poisson distribution as observations, we obtain the particular form of covariance as:*

$$\text{Cov}[R_{ij}, R_{tl}] = K[\delta_{it}(\mu_p\sigma_q)^2 + \delta_{jl}(\mu_q\sigma_p)^2 + \delta_{it}\delta_{jl}((\sigma_p\sigma_q)^2 + \mu_p\mu_q)]. \tag{48}$$

*Proof.*

- Recall the total law of covariance (Equation (33)):

$$\text{Cov}[R_{ij}, R_{tl}] = \mathbb{E}[Cov[R_{ij}, R_{tl}|p_i, q_j, p_t, q_l]] + \text{Cov}[\mathbb{E}[R_{ij}|p_i, q_j], \mathbb{E}[R_{tl}|p_t, ql]]$$

$$= \mathbb{E}[\delta_{it}\delta_{jl}\mathbb{V}[R_{ij}|p_i, q_j]] + \text{Cov}\left[\sum_k p_{ik}q_{jk}, \sum_k p_{tk}q_{lk}\right]$$

$$= \delta_{it}\delta_{jl}\mathbb{E}[\mathbb{V}[R_{ij}|P,Q]] + K[\delta_{it}(\mu_p\sigma_q)^2 + \delta_{jl}(\mu_q\sigma_p)^2 + \delta_{it}\delta_{jl}(\sigma_p\sigma_q)^2];$$

- Referring to Equation (46), we know that for Poisson distribution $\mathbb{E}[\mathbb{V}[R_{ij}|P,Q]] = \mathbb{E}[R_{ij}]$. Therefore Equation (48) holds.

$\square$

**Proposition 8.** *The correlation of true relevance with four different cases:*

$$\rho[R_{ij}, R_{tl}] = \begin{cases} 0, & \text{if} \quad i \neq t \quad \& \quad j \neq l \\ 1, & \text{if} \quad i = t \quad \& \quad j = l \\ \rho_1^* = \frac{K(\mu_q\sigma_p)^2}{\mathbb{V}[R_{ij}]}, & \text{if} \quad i = t \quad \& \quad j \neq l \\ \rho_2^* = \frac{K(\mu_p\sigma_q)^2}{\mathbb{V}[R_{ij}]}, & \text{if} \quad i \neq t \quad \& \quad j = l \end{cases} \tag{49}$$

$\rho_1^*$ *and* $\rho_2^*$ *refer to the correlation of true relevance* $R$, *in comparison with* $\rho_1$ *and* $\rho_2$ *that refer to observation* $Y$.

*Proof.* We know the relationship between the covariance and correlation $\rho[R_{ij}, R_{tl}] = \frac{Cov[R_{ij}, R_{tl}]}{\sqrt{\mathbb{V}[R_{ij}]\mathbb{V}[R_{tl}]}}$ and we analyze the four different cases that is similar to Proposition 5 so the above proposition holds:

1. *if   $i \neq t$   &   $j \neq l$*: due to the independence of these variables, we know that:
$$\rho[R_{ij}, R_{tl}] = 0;$$

2. *if   $i = t$   &   $j = l$*:
$$\rho[R_{ij}, R_{tl}] = \rho[R_{ij}, R_{ij}] = \frac{\text{Cov}[R_{ij}, R_{ij}]}{\sqrt{\mathbb{V}[R_{ij}]^2}} = 1;$$

3. *if   $i = t$   &   $j \neq l$*: this case indicates the row correlation
$$\rho[R_{ij}, R_{tl}] = \rho[R_{ij}, R_{il}] = \frac{\text{Cov}[R_{ij}, R_{il}]}{\sqrt{\mathbb{V}[R_{ij}]^2}} = \frac{K(\mu_q \sigma_p)^2}{\mathbb{V}[R_{ij}]};$$

4. *if   $i \neq t$   &   $j = l$*: this case indicates the column correlation
$$\rho[R_{ij}, R_{tl}] = \rho[R_{ij}, R_{tj}] = \frac{Cov[R_{ij}, R_{tj}]}{\sqrt{\mathbb{V}[R_{ij}]^2}} = \frac{K(\mu_p \sigma_q)^2}{\mathbb{V}[R_{ij}]}.$$

$\square$

## A.5   Determine the hyperparameters without exposure

In this section, we show how to determine the hyperparameters for generic model (Equation (5)) without exposure. In this case, we assume that no entries in the relevance matrix $R$ are missing.

**Proposition 9.** *Assume that the relevance matrix $R$ is dense. Given that we know $\mathbb{E}[R_{ij}]$, $\mathbb{V}[R_{ij}]$, $\rho_1^*$ and $\rho_2^*$, we can obtain the closed form for the true number of latent factor $K$ denoted as $K^*$ as:*

$$K^* = \frac{(1 - (\rho_1^* + \rho_2^*))\mathbb{V}[R_{ij}] - \mathbb{E}[\mathbb{V}[R_{ij}|P, Q]]}{\rho_1^* \rho_2^*} \left(\frac{\mathbb{E}[R_{ij}]}{\mathbb{V}[R_{ij}]}\right)^2. \tag{50}$$

*Proof.* In Proposition 8 we know that $\rho_1^* = \frac{K(\mu_q \sigma_p)^2}{\mathbb{V}[R_{ij}]}$, thus $K(\mu_q \sigma_p)^2 = \rho_1^* \mathbb{V}[R_{ij}]$. Similarly we have $K(\mu_p \sigma_q)^2 = \rho_2^* \mathbb{V}[R_{ij}]$. And we can further multiply the two terms:

$$\rho_1^* \rho_2^* \mathbb{V}[R_{ij}]^2 = K^2 (\mu_p \mu_q \sigma_p \sigma_q)^2 = \mathbb{E}[R_{ij}]^2 (\sigma_p \sigma_q)^2$$

$$\Rightarrow (\sigma_p \sigma_q)^2 = \rho_1^* \rho_2^* \left(\frac{\mathbb{V}[R_{ij}]}{\mathbb{E}[R_{ij}]}\right)^2.$$

Recall the variance of true relevance (Equation (44)):

$$\mathbb{V}[R_{ij}] = K[(\mu_p \sigma_q)^2 + (\mu_q \sigma_p)^2 + (\sigma_p \sigma_q)^2] + \mathbb{E}[\mathbb{V}[R_{ij}|P, Q]]$$

$$= (\rho_1^* + \rho_2^*)\mathbb{V}[R_{ij}] + K\rho_1^* \rho_2^* \left(\frac{\mathbb{V}[R_{ij}]}{\mathbb{E}[R_{ij}]}\right)^2 + \mathbb{E}[\mathbb{V}[R_{ij}|P, Q]].$$

Solve the above equation for $K$, we obtain the Equation( 50). $\square$

## A.6   Determine the hyperparameters with exposure

In this section, we show that how to determine the hyperparameters for our exposure-based model. The complete definition is described in Section 3.1.

**Proposition 10.** *For any pair of entries $Y_{ij}$ and $Y_{tl}$, the generic expansion of the two entries is given as follows:*

$$\begin{aligned}
\text{Cov}[Y_{ij}, Y_{tl}] &= \text{Cov}[O_{ij} R_{ij}, O_{tl} R_{tl}] \\
&= \text{Cov}[O_{ij}, O_{tl}]\text{Cov}[R_{ij}, R_{tl}] + \mathbb{E}[R_{ij}]\mathbb{E}[R_{tl}]\text{Cov}[O_{ij}, O_{tl}] + \mathbb{E}[O_{ij}]\mathbb{E}[O_{tl}]\text{Cov}[R_{ij}, R_{tl}]
\end{aligned} \tag{51}$$

*Proof.* By the Equation (38), the above equation holds. $\square$

### A.6.1 Constant exposure hyperparameter

In this case we consider the hyperparameter of the Bernoulli variable $O_{ij}$ as constant: $O_{ij} \sim Bern(\theta)$.

We first derive the estimator for Model 1:

**Proposition 11.** *Assume that the observation matrix $Y$ is sparse and random variable $O_{ij}$ and $O_{tl}$ follow exactly the same distribution $O_{ij}, O_{tl} \sim Bern(\theta)$. Given that we obtain sample of $\mathbb{E}[Y_{ij}]$, $\mathbb{V}[Y_{ij}]$, $\rho_1$ and $\rho_2$, we can obtain the estimator of number of latent factor $K$ denoted as $\hat{K}$ as:*

$$\hat{K} = \frac{(1 - (\rho_1^* + \rho_2^*))\mathbb{V}[R_{ij}] - \mathbb{E}[\mathbb{V}[R_{ij}|P,Q]]}{\rho_1^* \rho_2^*} \left(\frac{\mathbb{E}[R_{ij}]}{\mathbb{V}[R_{ij}]}\right)^2, \tag{52}$$

*where,*

$$\mathbb{E}[R_{ij}] = \theta^{-1}\mathbb{E}[Y_{ij}] \tag{53}$$

$$\mathbb{V}[R_{ij}] = \theta^{-1}\mathbb{V}[Y_{ij}] + \theta^{-2}(\theta - 1)\mathbb{E}[Y_{ij}]^2 \tag{54}$$

$$\rho^* = \frac{\mathbb{V}[Y_{ij}]\rho - \theta(1 - \theta)\mathbb{E}[R_{ij}]^2}{\theta\mathbb{V}[R_{ij}]}. \tag{55}$$

*Proof.* Because $O_{ij}$ and $O_{tl}$ follow exactly the same distribution, thus:

$$\mathbb{E}[O_{ij}] = \mathbb{E}[O_{tl}] = \theta,$$
$$\text{Cov}[O_{ij}, O_{tl}] = Var(\theta) = \theta(1 - \theta).$$

Recall Equation (10), we have:

$$\text{Cov}[Y_{ij}, Y_{tl}] = \theta(1 - \theta)\text{Cov}[R_{ij}, R_{tl}] + \theta(1 - \theta)\mathbb{E}[R_{ij}]\mathbb{E}[R_{tl}] + \theta^2\text{Cov}[R_{ij}, R_{tl}]$$
$$= \theta\text{Cov}[R_{ij}, R_{tl}] + \theta(1 - \theta)\mathbb{E}[R_{ij}]^2.$$

By the definition of covariance, we know that $\text{Cov}[Y_{ij}, Y_{tl}] = \mathbb{V}[Y_{ij}]\rho$ and $\text{Cov}[R_{ij}, R_{tl}] = \mathbb{V}[R_{ij}]\rho^*$, and we solve $\rho^*$ as Equation (55).

$\square$

We then derive the estimator for Model 2:

**Proposition 12.** *Assume that the observation matrix $Y$ is sparse and the random variable $O_{ij}$ and $O_{tl}$ are independent and identically distributed $O_{ij}, O_{tl} \overset{\text{iid}}{\sim} Bern(\theta)$. Given that we obtain sample of $\mathbb{E}[Y_{ij}]$, $\mathbb{V}[Y_{ij}]$, $\rho_1$ and $\rho_2$, we can obtain the estimator of number of latent factor $K$ denoted as $\hat{K}$ as:*

$$\hat{K} = \frac{(1 - (\rho_1^* + \rho_2^*))\mathbb{V}[R_{ij}] - \mathbb{E}[\mathbb{V}[R_{ij}|P,Q]]}{\rho_1^* \rho_2^*} \left(\frac{\mathbb{E}[R_{ij}]}{\mathbb{V}[R_{ij}]}\right)^2, \tag{56}$$

*where,*

$$\mathbb{E}[R_{ij}] = \theta^{-1}\mathbb{E}[Y_{ij}] \tag{57}$$

$$\mathbb{V}[R_{ij}] = \theta^{-1}\mathbb{V}[Y_{ij}] + \theta^{-2}(\theta - 1)\mathbb{E}[Y_{ij}]^2 \tag{58}$$

$$\rho^* = \theta^{-2}\left(\frac{\mathbb{V}[Y_{ij}]}{\mathbb{V}[R_{ij}]}\right)\rho. \tag{59}$$

*Proof.* Because $O_{ij}$ and $O_{tl}$ are independent and identically distributed, thus:

$$\mathbb{E}[O_{ij}] = \mathbb{E}[O_{tl}] = \theta,$$
$$\text{Cov}[O_{ij}, O_{tl}] = 0.$$

Recall Equation (10), we have:

$$\mathrm{Cov}[Y_{ij}, Y_{tl}] = \theta^2 \mathrm{Cov}[R_{ij}, R_{tl}].$$

By the definition of covariance, we know that $\mathrm{Cov}[Y_{ij}, Y_{tl}] = \mathbb{V}[Y_{ij}]\rho$ and $\mathrm{Cov}[R_{ij}, R_{tl}] = \mathbb{V}[R_{ij}]\rho^*$, and we solve $\rho^*$ as Equation (59).

$\square$

### A.7 Correlated Latent Features

In this section, we derive the results described in Section 5.3.

We know that:

$$\mathrm{Cov}[R_{ij}, R_{tl}] = \mathrm{Cov}\left[\sum_k p_{ik} q_{jk}, \sum_k p_{tk} q_{lk}\right]$$
$$= \sum_{k,k'} \mathrm{Cov}[p_{ik} q_{jk}, p_{tk'} q_{lk'}].$$

For the covariance in the same row that $if \quad i = t \quad \& \quad j \neq l$:

$$\sum_{k,k'} \mathrm{Cov}[p_{ik} q_{jk}, p_{tk'} q_{lk'}] = \sum_{k,k'} \mathrm{Cov}[p_{ik} q_{jk}, p_{ik'} q_{lk'}]$$
$$= \sum_k \mathrm{Cov}[p_{ik} q_{jk}, p_{ik} q_{lk}] + \sum_{k \neq k'} \mathrm{Cov}[p_{ik} q_{jk}, p_{ik'} q_{lk'}].$$

We know the result of the first term when $k \neq k'$, now we calculate the second term when $k \neq k'$. We first calculate the inside term:

$$\mathrm{Cov}[p_{ik} q_{jk}, p_{ik'} q_{lk'}] = \mathbb{E}[p_{ik} p_{ik'} q_{jk} q_{lk'}] - \mathbb{E}[p_{ik} q_{jk}]\mathbb{E}[p_{ik'} q_{lk'}]$$
$$= \mathbb{E}[p_{ik} p_{ik'}]\mathbb{E}[q_{jk} q_{lk'}] - \mathbb{E}[p_{ik}]\mathbb{E}[q_{jk}]\mathbb{E}[p_{ik'}]\mathbb{E}[q_{lk'}]$$
$$= (\mathbb{E}[p_{ik}]\mathbb{E}[p_{ik'}] + \mathrm{Cov}[p_{ik}, p_{ik'}])(\mathbb{E}[q_{jk}]\mathbb{E}[q_{lk'}] + \underbrace{\mathrm{Cov}[q_{jk}, q_{lk'}]}_{=0}))$$
$$- (\mathbb{E}[p_{ik}]\mathbb{E}[q_{jk}]\mathbb{E}[p_{ik'}]\mathbb{E}[q_{lk'}])$$
$$= \mathbb{E}[q_{jk}]\mathbb{E}[q_{lk'}]\mathrm{Cov}[p_{ik}, p_{ik'}].$$

Then we have:

$$\mathrm{Cov}[R_{ij}, R_{il}] = \sum_k \mathbb{E}[q_{jk}]\mathbb{E}[q_{lk}]\mathbb{V}[p_{ik}] + \sum_{k \neq k'} \mathbb{E}[q_{jk}]\mathbb{E}[q_{lk'}]\mathrm{Cov}[p_{ik}, p_{ik'}].$$

Assume latent factors follow Multivariate Distribution denoted as $\mathcal{F}_k$:

$$p_i \sim \mathcal{F}_k(\mu_p, \Sigma_p), \quad q_j \sim \mathcal{F}_k(\mu_q, \Sigma_q), \tag{60}$$

finally, we obtain the closed form of the covariance,

$$\mathrm{Cov}[R_{ij}, R_{il}] = \sum_k \mathbb{E}[q_{jk}]\mathbb{E}[q_{lk}]\mathbb{V}[p_{ik}] + \sum_{k \neq k'} \mathbb{E}[q_{jk}]\mathbb{E}[q_{lk'}]\mathrm{Cov}[p_{ik}, p_{ik'}]$$
$$= \mu_q^2 \underbrace{\left(\sum_k \mathbb{V}[p_{ik}] + \sum_{k \neq k'} \mathrm{Cov}[p_{ik}, p_{ik'}]\right)}_{\textbf{The element-wise sum of } \Sigma_p}$$
$$= \mu_q^2 (\mathbf{1}^T \Sigma_p \mathbf{1}).$$

The above notation refers to the all-ones vector. However, the covariance matrix is always positive semi-definite, and the covariance of any two entries is always non-negative.

## B   Additional Figures & Tables

The following tow figures are illustrations of the two different generic Bayesian MF models.

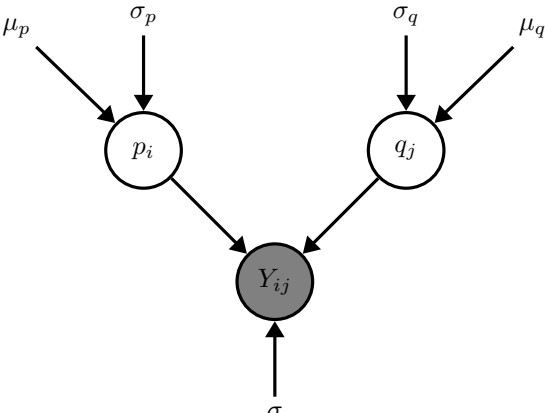

Figure 5: Illustration of Generic Bayesian MF

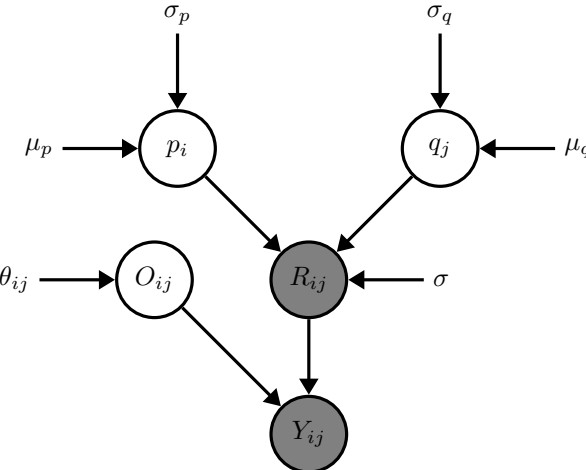

Figure 6: Illustration of Generic Exposure-based Bayesian MF

The following table shows all of the hyperparameters initialization described in the previous research (da Silva et al., 2023).

Table 4: Hyperparameters Initialization for Different Specifications

| Spec. | $a$ | $b$ | $c$ | $d$ | $\mu_p$ | $\sigma_p$ | $\mu_q$ | $\sigma_q$ | $\mathbb{E}[R]$ | $\mathbb{V}[R]$ |
|-------|-----|-----|-----|-----|---------|------------|---------|------------|-----------------|-----------------|
| A | 10 | 1 | 10 | 1 | 10.0 | 3.16 | 10.0 | 3.16 | 2500.00 | 55000.00 |
| B | 10 | 2 | 10 | 2 | 5.0 | 1.58 | 5.0 | 1.58 | 625.00 | 3906.25 |
| C | 0.001 | 0.01 | 0.01 | 0.1 | 0.1 | 3.16 | 0.1 | 1.0 | 0.25 | 253.00 |
| D | 0.1 | 1 | 0.1 | 1 | 0.1 | 0.32 | 0.1 | 0.32 | 0.25 | 0.55 |
| E | 0.1 | 0.1 | 0.1 | 0.1 | 1.0 | 3.16 | 1.0 | 3.16 | 25.00 | 3025.00 |
| F | 1 | 1 | 0.1 | 0.1 | 1.0 | 1.0 | 1.0 | 3.16 | 25.00 | 550.00 |
| G | 1000 | 1000 | 1000 | 1000 | 1.0 | 0.03 | 1.0 | 0.03 | 25.00 | 25.05 |

## C   Additional Experiments on Real-World Datasets

We conducted additional experiments on real-world datasets MovieLens (Harper & Konstan, 2015), which has been widely studied for recommender systems. The datasets contain users' ratings of different movies on a 5-star scale, with half-star increments (0.5 stars - 5.0 stars). We selected three MovieLens datasets with different sizes, from 100k records to 10m records. The following table shows the basic information and corresponding results of each dataset.

Table 5: Information of each MovieLens datasets and the corresponding estimation of the number of latent factor K by the different models

| *dataset* | *num_users* | *num_movies* | *num_ratings* | *sparsity* | *mean* | *var* | *ori_K* | *m1_K* | *m2_K* |
|-----------|-------------|--------------|---------------|------------|--------|-------|---------|--------|--------|
| **ml-100k** | 943 | 1682 | 100k | 0.94 | 3.53 | 1.27 | 389.81 | 36.95 | 24224.29 |
| **ml-1m** | 6040 | 3706 | 1m | 0.96 | 3.58 | 1.25 | 504.39 | 39.58 | 37499.15 |
| **ml-10m** | 69878 | 10677 | 10m | 0.99 | 3.51 | 1.12 | 614.01 | 45.17 | 58315.81 |

The **num_users** and **num_movies** denote the total number of users and movies in each dataset. The **num_ratings** represents the total number of ratings or records in each dataset, ranging from 100,000 to 10 million. The **sparsity** refers to the ratio of the total number of ratings to the total number of entries in the rating matrix. In other words, the parameter of the Bernoulli variable used in our paper, denoted as $\theta$ (Pobs), equals $1 - sparsity$. The **mean** and **var** signify the mean and variance sampled from the dataset. The **ori_K**, **m1_K**, and **m2_K** represent the estimated number of latent factors K for the original model, our proposed Model 1, and Model 2, respectively. Given that the optimal K value is influenced by various factors, such as the selection of the factorization model and the characteristics of the dataset, it is challenging to identify the optimal K value as a benchmark. Based on different matrix factorization experiments, a reasonable K value is typically selected between 50 and 200. Our additional results demonstrate that our Model 1 generates the most comparable **m1_K** values to the most commonly chosen K value. Conversely, the exceptionally large **m2_K** values suggest that our Model 2 may not be adequately suited to these MovieLens datasets. In summary, our proposed Model 1 provides more appropriate estimations of K compared to the original model, particularly for these MovieLens datasets with varying sizes and sparsities.

We measured the execution time of three different models on both our synthetic and the real-world datasets mentioned above.

Table 6: Execution time (in seconds) of the three different models on our synthetic an the MovieLens datasets

| *dataset* | *ori (s)* | *m1 (s)* | *m2 (s)* | *sampling (s)* |
|-----------|-----------|----------|----------|----------------|
| **synthetic** | $8.11 * 10^{-6}$ | $7.15 * 10^{-6}$ | $4.05 * 10^{-6}$ | 0.01 |
| **ml-100k** | $9.06 * 10^{-6}$ | $6.91 * 10^{-6}$ | $4.77 * 10^{-6}$ | 0.56 |
| **ml-1m** | $1.29 * 10^{-5}$ | $6.19 * 10^{-6}$ | $3.81 * 10^{-6}$ | 1.10 |
| **ml-10m** | $5.61 * 10^{-5}$ | $2.79 * 10^{-5}$ | $6.91 * 10^{-6}$ | 211.19 |

In Table 6, the ***ori***, ***m1***, and ***m2*** represent a single estimation time for the original model, our proposed Model 1, and Model 2, respectively. It shows that the estimation times for our synthetic datasets and real-world datasets are within a minuscule range and exhibit no significant difference.

In fact, the most computationally expensive aspect is the sampling process of the correlation coefficients shared by all models. The ***sampling*** column represents the sampling time of datasets with varying scales. The sampling time increases exponentially as the dataset size increases, accounting for the majority of the total execution time.

