# OpenReview forum: "Prior Specification for Exposure-based Bayesian Matrix Factorization"
_TMLR — Accepted by TMLR_

### Review · Reviewer_xU9C · 2025-05-30

**Summary Of Contributions:**

Authors consider recommender systems (RSs).They analyze how Prior Predictive Matching (PPM) which has been used before performs as as data sparsity increases. They present an enhanced method for specifying priors in Bayesian matrix factorization models and improve the estimators by implementing an exposure-based model to better simulate data scarcity.
Their method demonstrates accuracy improvements in hyperparameter estimation during synthetic experiments.
They also explore the feasibilityof applying this method to real-world datasets and provide insights into how the model’s behavior adapts to varying levels of data sparsity.

They are making three assumptions on the model that uses exposure variable to simulate data sprsity/missing values:

1) the exposure variable $O_{ij}$ is independent of the truerelevance $R_{ij} $

2) the observation model $F_R$ is Poisson distribution

3a)  the random variable $O_{ij}$ and $O_{tl}$ follow exactly the same distribution $O_{ij}, O_{tl} ∼ Bern(\theta)$ and $Cov(O_{ij}, O_{tl})=Var(\theta)=\theta(1-\theta)$. (Model 1) or

3b) the random variable $O_{ij}$ and $O_{tl}$ are independent and identically distributed,$O_{ij}, O_{tl} ∼ Bern(\theta)$ (Model 2)

Authors justify their models by stating theoretical propositions and model equations with proofs in the appendix.

**Audience:**

Yes

**Claims And Evidence:**

Yes

**Requested Changes:**

Is there also a difference in running times between the different models?

**Strengths And Weaknesses:**

Strengths:

- The write up is well done and clear. The problem considered seems intersting even though I can't say how important missing values are in recommender systems as I'm not really familiar with them.

- The paper discuessed theory and there expermients on synthetic data and it mentions both adavantages but also limitations of their methods.

Weaknesses:

- their model only perfoms better for sparse data.

---

> ### Author Response · Authors · 2025-06-29
>
> Dear Reviewer, thank you very much for your sincere comments. We would like to answer your current question as the following:
>
> ## Q1. Execution time difference
> - The following table shows the execution time of the three models in a single estimation. Notably, all execution times are within a minuscule range and have no significant difference.
>
> | **_Models_** | **_Times (s)_** |
> |--------------|-----------------|
> | **Original** | $8.11*10^{-6}$  |
> | **Model 1**  | $7.15*10^{-6}$  |
> | **Model 2**  | $4.05*10^{-6}$  |
>
> - Because we conducted additional experiments on the real-world datasets, the MovieLens, we also compared the model estimation time with differernt data size. The following table contains the basic information of these datasets and our observed excuation times.
>
> | **_dataset_** | **_num_users_** | **_num_movies_** | **_num_ratings_** | **_sparsity_** | **_mean_** | **_var_** | **_ori_ (s)**  | **_m1_ (s)**   | **_m2_ (s)**   | **_sampling_  (s)** |
> |---------------|-----------------|------------------|-------------------|----------------|------------|-----------|----------------|----------------|----------------|----------------------|
> | **ml-100k**   | 943             | 1682             | 100k              | 0.94           | 3.53       | 1.27      | $9.06*10^{-6}$ | $6.91*10^{-6}$ | $4.77*10^{-6}$ | $0.56$               |
> | **ml-1m**     | 6040            | 3706             | 1m                | 0.96           | 3.58       | 1.25      | $1.29*10^{-5}$ | $6.19*10^{-6}$ | $3.81*10^{-6}$ | $1.10$               |
> | **ml-10m**    | 69878           | 10677            | 10m               | 0.99           | 3.51       | 1.12      | $5.61*10^{-5}$ | $2.79*10^{-5}$ | $6.91*10^{-6}$ | $211.19$             |
>
> The datasets contain users' ratings of different movies on a 5-star scale, with half-star increments (0.5 stars - 5.0 stars). We selected 3 MovieLens datasets with different sizes, from 100k records to 10m records. The “_num_users_” and “_num_movies_” denote the total number of users and movies in each dataset. The “_num_ratings_” represents the total number of ratings or records in each dataset, ranging from 100,000 to 10 million. The “_sparsity_” refers to the ratio of the total number of ratings to the total number of entries in the rating matrix. In other words, the parameter of the Bernoulli variable used in our paper, denoted as $\theta$ (_Pobs_), equals $1-\textit{sparsity}$. The “_mean_” and “_var_” signify the mean and variance sampled from the dataset.
>
> The “_ori_”, “_m1_”, and “_m2_” represent a single estimation time for the original model, our proposed Model 1, and Model 2, respectively. It shows that even on real-world datasets, the estimation times are within a minuscule range and have no significant difference.
>
> In fact, the most computationally expensive aspect is the sampling process of the correlation coefficients shared by all models. The “_sampling_” column represents the sampling time of datasets with varying scales. The sampling time increases exponentially as the dataset size increases, accounting for the majority of the total execution time.

---

### Review · Reviewer_Ck3Y · 2025-06-06

**Summary Of Contributions:**

This paper proposes an enhanced prior specification method for Bayesian Matrix Factorization (MF) that integrates exposure-based modeling to improve hyperparameter estimation in the presence of increased data sparsity. The paper builds upon the Prior Predictive Matching (PPM) framework, which aligns prior distributions with observed data to estimate hyperparameters without costly posterior inference. While PPM performs well in dense datasets, its performance deteriorates significantly as sparsity increases—a common characteristic in real-world recommender systems. To address this, the paper introduces an exposure-based Bayesian MF model that accounts for missing-not-at-random (MNAR) data mechanisms by incorporating a Bernoulli exposure variable into the probabilistic framework. This allows the model to distinguish between true relevance (latent interaction) and observed data, better reflecting implicit feedback behavior. Two estimators for the number of latent factors are derived under different assumptions of exposure variable correlation: one assumes all exposure variables share the same distribution, and the other assumes they are i.i.d. These estimators utilize expectations, variances, and correlation statistics of the sparse observation matrix to infer model hyperparameters. The method is empirically evaluated on synthetic datasets across different sparsity levels and prior specifications, comparing the original PPM-based estimator with the new exposure-based versions. This approach is tested empirically using two models and compared with various baseline models.

**Audience:**

Yes

**Broader Impact Concerns:**

There is no broader impact statement, but the paper can be used in areas beyond recommender systems. A paragraph could be added to discuss potential impacts.

**Claims And Evidence:**

Yes

**Requested Changes:**

I would suggest working on the weaknesses mentioned above. Specifically, it would be interesting to see how the proposed exposure-based estimator perform on real-world recommendation datasets (e.g., MovieLens, Amazon, or Netflix), particularly those with high sparsity and implicit feedback. I know that the synthetic experiments aim to have a controlled setting for known conditions of the priors. However, it will be good to verify the applicability to a couple of real scenarios.

A controlled analysis that could also clarify the utility of the model is to test how sensitive the proposed estimators are to dataset characteristics such as the user-item interaction distribution or popularity bias.

Finally, a minor comment regarding the readability issue mentioned above is to ensure that variables are introduced and discussed when they are used.

I also have a few questions.

How robust are the estimators to misspecification of the exposure model (e.g., if the true missingness mechanism is not Bernoulli)?

Under what conditions do the proposed estimators break down or yield unstable estimates?

**Strengths And Weaknesses:**

The main strength of this paper is that it clearly articulates the challenge of prior specification in Bayesian Matrix Factorization under data sparsity, grounding it in real-world issues common to recommender systems. It effectively motivates the need for an exposure-based refinement of existing methods.
Another strength is that the synthetic experiments systematically vary sparsity, latent dimensions, and prior configurations to stress-test the estimators. The inclusion of baseline comparisons and quantile-based error plots contributes to a solid and reproducible evaluation framework.

The main weakness is that, although the paper is framed around practical recommender system applications, all experiments are conducted on synthetic datasets. The lack of experiments on real-world data limits insights into practical deployment and generalization performance.
From a presentation perspective, some parts of the paper—especially in the theoretical and experimental sections—suffer from verbose text and dense notations. This may hinder readability and accessibility for a broader audience. For instance, while the paper refers to $\rho$ and $\rho*$ in Equation (5), Equations (17) and (19) refer to $\rho*$ but no clarification is provided and only further reading, including the appendix, clarifies this. Another important weakness is that the paper underemphasizes potential limitations of the proposed approach (e.g., sensitivity to the choice of exposure model, assumptions of independence or correlation), and does not compare against alternative methods beyond da Silva et al. (2023).

---

> ### Author Response · Authors · 2025-06-29
>
> Dear Reviewer,
> thank you very much for your sincere comments. We would like to answer your current questions in the following manner:
>
> ## Q1. Additional experiments on real-world datasets
> We conducted additional experiments on real-world datasets MovieLens, which has been wildly studied for recommender systems. The datasets contain users' ratings of different movies on a 5-star scale, with half-star increments (0.5 stars - 5.0 stars). We selected 3 MovieLens datasets with different sizes, from 100k records to 10m records. The following table shows the basic information and corresponding results of each dataset.
>
> | **_dataset_** | **_num_users_** | **_num_movies_** | **_num_ratings_** | **_sparsity_** | **_mean_** | **_var_** | **_ori_K_** | **_m1_K_** | **_m2_K_** |
> |---------------|-----------------|------------------|-------------------|----------------|------------|-----------|-------------|------------|------------|
> | **ml-100k**   | 943             | 1682             | 100k              | 0.94           | 3.53       | 1.27      | 389.81      | 36.95      | 24224.29   |
> | **ml-1m**     | 6040            | 3706             | 1m                | 0.96           | 3.58       | 1.25      | 504.39      | 39.58      | 37499.15   |
> | **ml-10m**    | 69878           | 10677            | 10m               | 0.99           | 3.51       | 1.12      | 614.01      | 45.17      | 58315.81   |
>
> The “_num_users_” and “_num_movies_” denote the total number of users and movies in each dataset. The “_num_ratings_” represents the total number of ratings or records in each dataset, ranging from 100,000 to 10 million. The “_sparsity_” refers to the ratio of the total number of ratings to the total number of entries in the rating matrix. In other words, the parameter of the Bernoulli variable used in our paper, denoted as $\theta$ (_Pobs_), equals $1-\textit{sparsity}$. The “_mean_” and “_var_” signify the mean and variance sampled from the dataset.
> The “_ori_K_”, “_m1_K_”, and “_m2_K_” represent the estimated number of latent factors K for the original model, our proposed Model 1, and Model 2, respectively. Given that the optimal K value is influenced by various factors, such as the selection of the factorization model and the characteristics of the dataset, it is challenging to identify the optimal K value as a benchmark. Based on different matrix factorization experiments, a reasonable K value is typically selected between 50 and 200. Our additional results demonstrate that our Model 1 generates the most comparable “_m1_K_” values to the most commonly chosen K value. Conversely, the exceptionally large “_m2_K_” values suggest that our Model 2 may not be adequately suited to these MovieLens datasets. In summary, our proposed Model 1 provides more appropriate estimations of K compared to the original model, particularly for these MovieLens datasets with varying sizes and sparsities.
>
> We will add the additional results to our appendix.
>
> ## Q2. Readability issues of notations
>
> For the specific confusing notations about $\rho$ and $\rho^\*$, we would like to clarify as follows: the $\rho$ denotes the original correlation coefficients sampled from the dataset. The $\rho^\*$ denotes the modified $\rho$ in our different models. The $\rho^\*_1$ and $\rho^\*_2$ denote the raw/column $\rho^\*$ respectively.
>
> We will add the description to Equation(15).

---

> ### Author Response · Authors · 2025-06-29
>
> ## Q3. Robustness of the estimators to misspecification
> In our synthetic dataset, we generated the exposure variables by sampling from a binomial distribution. This suggests that Model 2, with the independent and identically distributed (i.i.d.) assumption, should be more adaptable to the synthetic dataset than Model 1. Nevertheless, both models can yield satisfactory outcomes when the specification changes.
>
> Referring to the MovieLens datasets we used in Q1, these real-world datasets are not necessarily missing in a Bernoulli way. However, the Model 1 still generates appropriate estimations with reasonable value increasing as the datasets expand in size. This observation suggests that the robustness of our estimators is contingent upon the specific datasets employed.
>
> ## Q4. Conditions of unstable estimations
> In fact, we observed unstable relative errors in our synthetic experiments. These relative errors generally increase to exceptionally large values at some point, then decrease and become stable, while the datasets become sparse ($\theta$ decreases).
>
> In order to analyze the behavior of the relative error, we focus on the term $\frac{\hat{K}}{K}$. The $\hat{K}$ denotes the estimated value, and $K$ denotes the true value. The $\mathbb{E}[Y]$ denotes the true expectation and the $\mathbb{V}[Y]$ denotes the true variance. For simplicity of the equation, we use the following notation:
> $$
> \tau = 1+(1-\theta^{-1})(\frac{\mathbb{E}[Y]^2}{\mathbb{V}[Y]}).
> $$
>
> - For Model 1:
> $$
>     \frac{\hat{K}}{K} = \frac{\rho_1\rho_2}{(\rho_1-1+\tau)(\rho_2-1+\tau)} \frac{(2-\tau-(\rho_1+\rho_2))-\frac{\mathbb{E}[Y]}{\mathbb{V}[Y]}}{(1-(\rho_1+\rho_2))-\frac{\mathbb{E}[Y]}{\mathbb{V}[Y]}}.
> $$ We have observed that the above term may encounter a zero denominator, resulting in an exceptionally large relative error. Although the sampled correlation coefficients are limited in range $[-1, 1]$, the sampling accuracy becomes unreliable while $\theta$ decreases. Therefore, we can represent the "singular point" $\theta_0$, which may cause a zero denominator as follows($\rho_1$ is equivalent to $\rho_2$ in the equation):
> $$\theta_0 =\frac{1}{1+(\frac{\mathbb{V}[Y]}{\mathbb{E}[Y]^2})\rho_1}.$$
> We noticed that the term $\frac{\mathbb{V}[Y]}{\mathbb{E}[Y]^2}$ is constant for each specification. As a result, the inaccurate sampling of correlation coefficients for a particular dataset can lead to unstable estimation at varying sparsity levels.
>
> - For Model 2:
> $$
>     \frac{\hat{K}}{K} = \frac{(\tau-\theta^{-1}(\rho_1+\rho_2))-\frac{\mathbb{E}[Y]}{\mathbb{V}[Y]}}{(1-(\rho_1+\rho_2))-\frac{\mathbb{E}[Y]}{\mathbb{V}[Y]}}\theta.
> $$ Different from the ratio form of Model 1, the above term only has one denominator independent of variable $\theta$, which means Model 2 is less likely to have a zero denominator.
>
> In summary, the probability of unstable estimation is affected by the specific datasets and the sampling quality of correlation coefficients. Please refer to Section 5.1 for more details.

---

### Review · Reviewer_yKEJ · 2025-08-04

**Summary Of Contributions:**

- This paper investigates how Prior Predictive Matching struggles to set good priors in Bayesian matrix factorization when data is sparse.
- To fix that, the authors propose a new model that adds an "exposure" variable, capturing whether a user even had the chance to interact with an item.
- They use this exposure-aware model to build better estimators for key hyperparameters like the number of latent factors.
- In experiments with synthetic data, their method consistently beats the original PPM approach, especially when lots of data is missing.
- Overall, they show that modeling exposure helps make prior specification more accurate and reliable in realistic recommender system settings.

**Audience:**

Yes

**Broader Impact Concerns:**

There are no broader impact concerns.

**Claims And Evidence:**

No

**Requested Changes:**

- It is difficult to analyze Figures 1, 2, and 3 and Table 3; please elaborate on these.
- I’m not sure why real-world datasets weren’t included, assuming I understood the setup correctly.
- The approach assumes exposure probabilities are known, but these are usually unobserved and tricky to estimate in practice.
- The model adds complexity, but there’s no discussion about how much computation it requires.
- The tables use vertical bars, which isn’t typical for academic formatting.
- Figure 4 has a rounded box around it.
- The figures don’t have grid lines, which would make the plots easier to read and compare visually.
- In Equation (15), the word "if" should be in regular text, not italic, since it’s not a variable.
- What are M1 and M2 in Model 1 and Model 2, respectively?
- A space is missing after (1), (2), and (3) in Section 4.2.
- Parentheses are missing in the references to equations throughout the manuscript.
- Figure 1 appears after Figures 2 and 3, which should be reordered.

**Strengths And Weaknesses:**

### Strengths

- The paper addresses a practical and well-motivated problem in Bayesian recommender systems, focusing on improving prior specification under sparse data.
- It introduces an exposure-based modeling approach that aligns with how missing data typically arises in real-world recommendation settings.
- The proposed method is supported by detailed theoretical derivations and shows consistent improvements in estimation accuracy in controlled synthetic experiments.

### Weaknesses

- The evaluation is limited to synthetic data.
- The approach assumes knowledge of exposure probabilities, which are generally unobserved.
- The method is primarily evaluated on estimating the number of latent factors, without broader analysis of downstream performance.
- The added modeling complexity is not accompanied by discussion of computational overhead.

---

> ### Author Response · Authors · 2025-08-18
>
> Dear Reviewer, thank you very much for your sincere comments. We would like to answer your current questions in the following manner:
>
> ## Q1. Additional experiments on real-world datasets
> Since the true distributions and exposure probabilities of real-world datasets are often unknown, we primarily focused on synthetic datasets in the main content for flexible and controlled analysis. This choice allowed us to compare our estimations with the actual prior settings.
> In the revised version, we conducted additional experiments on real-world datasets (MovieLens), which show that our proposed model can provide more appropriate estimations of K compared to the original model, particularly for these MovieLens datasets with varying sizes and sparsities.
>
> Please check our response to the previous reviewer and Appendix C in the revised manuscript for details.
>
> ## Q2. Unknown exposure probabilities
> We understand that the actual exposure probabilities are usually unobserved and very difficult to estimate in the first place. In the industrial scenarios, the exposure mechanisms can be very complicated, including the fundamental popularity-based and user-item profile-based methods, etc. Therefore, we developed our models with the fundamental exposure assumption to obtain theoretical results, the closed form of estimators.
> Referring to our additional results in Appendix C, it’s important to note that these real-world datasets may not be missing in a Bernoulli way. Nevertheless, our approaches with a determined exposure assumption can still yield satisfactory results, which also shows the robustness of our models to a certain extent.
>
> ## Q3. Broader analysis of downstream performance
> We focused on estimation of the number of latent factors $K$ for several reasons:
> - The number of latent factors $K$ exists in various types of matrix factorization models, including conventional and probabilistic methods. It is one of the most significant hyperparameters in matrix factorization models.
> - In our probabilistic model, the other hyperparameters (priors of the Gamma distribution) can be derived from the number of latent factors. Thus, the estimation accuracy of $K$ reflects the estimation accuracy of other hyperparameters.
> - The other downstream performances, such as recommendation accuracies, are highly dependent on the specific matrix factorization model and datasets. Since our primary focus is prior specification, we did not include such analysis in the main content.
>
> ## Q4. Computational overhead (Execution time difference)
> Our additional experiment results show that the estimation times for our synthetic datasets and real-world datasets are within a minuscule range and exhibit no significant difference. The most computationally expensive aspect is the sampling process of the correlation coefficients shared by all models, determined by the size of the datasets.
> Please check our response to the previous reviewer and Appendix C in the revised manuscript for details.
>
> ## Q5. Other issues
> We revised our manuscript according to the other suggestions:
> -   We optimized the visualizations of Figures 1, 2, 3, and 4.
> -   We reformatted Tables 1, 2, and 3.
> -   We revised inappropriate italic text in Equations (15) and (49).
> -   We added proper parentheses to all the equation references.
> -   M1 and M2 are abbreviations for Model 1 and Model 2.

---

### Decision · Action_Editor_zadY · 2025-09-07

**Recommendation:** Accept with minor revision

**Additional Comments:**

The paper is well written in general, but I would like to suggest a few minor things.
- Regarding Fig. 6 where the exposure-based BMF is illustrated, the arrows from both the exposure variable $O_{ij}$ and the
true relevance variable $R_{ij}$ should point directly to the final observation $Y_{ij}$. The current illustration does not properly reflects the relation $Y_{ij} = O_{ij} R_{ij}$.
- In my first reading, it was not easy to catch the contribution of the proposed model, compared to the existing exposure-based model briefly reviewed in Sec. 2.2. It would be nice to emphasize how different the proposed model is from the existing one, in very friendly manner.
​

**Audience:**

Yes

**Audience Explanation:**

The exposure-based model is one promising way to handle the issue of missing-not-at-random (MNAR). The method well addresses the problem of PPM when the data becomes  sparser, which is the case for real-world recommendation tasks. The method is sound ad enhances the existing work in a nice way, The approach in this paper will be of interest for practitioners working in recommender systems.

**Claims And Evidence:**

Yes

**Claims Explanation:**

This paper presents an enhanced prior specification method for Bayesian matrix factorization that integrates the generic probabilistic matrix factorization with an additional exposure component to simulate the scenario of missing data. It explicitly separates the underlying preference from the observation process, introducing  the hypothetical dense matrix $R_{ij}$ and the exposure random variable $O_{ij}$.
Unlike the previous exposure-based model, the proposed model allows for analytically deriving hyperparameter estimators that correct for sparsity.  The method is tested empirically using two models and compared with various baseline models. Synthetic experiments systematically vary sparsity, latent dimensions, and prior configurations to stress-test the estimators. Upon request by reviewers, the new experiments on real-world datasets were carried out.